# From Logits to Hierarchies: Hierarchical Clustering made Simple

Emanuele Palumbo [1 2]  Moritz Vandenhirtz [2]  Alain Ryser [2]  Imant Daunhawer[⋆ 2]  Julia E. Vogt[⋆ 2]

## Abstract

The hierarchical structure inherent in many real-world datasets makes the modeling of such hierarchies a crucial objective in both unsupervised and supervised machine learning. While recent advancements have introduced deep architectures specifically designed for hierarchical clustering, we adopt a critical perspective on this line of research. Our findings reveal that these methods face significant limitations in scalability and performance when applied to realistic datasets. Given these findings, we present an alternative approach and introduce a lightweight method that builds on pre-trained non-hierarchical clustering models. Remarkably, our approach outperforms specialized deep models for hierarchical clustering, and it is broadly applicable to any pre-trained clustering model that outputs logits, without requiring any fine-tuning. To highlight the generality of our approach, we extend its application to a supervised setting, demonstrating its ability to recover meaningful hierarchies from a pre-trained ImageNet classifier. Our results establish a practical and effective alternative to existing deep hierarchical clustering methods, with significant advantages in efficiency, scalability and performance.

## 1. Introduction

Modeling hierarchical structures in the data is a long-standing goal in machine learning research (Bengio et al., 2013; Jordan & Mitchell, 2015). In many real-world scenarios, data is inherently organized in hierarchies, such as phylogenetic trees (Linnæus, 1758; Sneath & Sokal, 1962; Penny, 2004), tumor subclasses (Sørlie et al., 2001) and social networks (Ravasz & Barabási, 2003; Crockett et al., 2017). In unsupervised learning, hierarchical clustering can provide more accurate insights than flat (i.e., non-hierarchical) clustering methods by introducing multiple levels of granularity and alleviating the need for a fixed number of clusters specified a priori (Bertsimas et al., 2021; Chami et al., 2020). This aids scientific understanding and interpretability by providing a more informative representation (Lipton, 2018; Marcinkevičs & Vogt, 2020). The benefits of modeling hierarchies in the data extend to supervised scenarios. For example, interpretable methods based on decision trees (Breiman, 2001; Tanno et al., 2019) hierarchically partition the data so that points in each split are linearly separable into classes. More recent work leverages hierarchies in the data to improve supervised methods (Bertinetto et al., 2020; Goren et al., 2024; Karthik et al., 2021) or for self-supervision (Long & van Noord, 2023).

Among classic algorithms for hierarchical clustering, agglomerative methods have been the most widely adopted. These methods compute pairwise distances between data points, often in a lower-dimensional representation space. Starting from the instance level, a hierarchy is then built based on the pairwise distances by recursive agglomeration of similar points or clusters together in a bottom-up fashion (Murtagh & Contreras, 2011). More recently, a revived interest in hierarchical clustering has sparked novel, sophisticated approaches using deep architectures (Mautz et al., 2020; Goyal et al., 2017; Shin et al., 2019; Vikram et al., 2019; Manduchi et al., 2023). These approaches require specialized architectures and complex training schemes. In this work, we uncover two key limitations of these approaches. First, we find that these methods struggle to handle large-scale datasets, failing to deliver satisfactory performance. This is in part due to their high computational demands and in part due to the difficulty in modeling a large number of clusters. Second, we observe a notable gap in performance at the leaf level compared to non-hierarchical models. This is particularly problematic, as the advantage of introducing a hierarchy should not come at the expense of the quality of clustering at the leaf-level granularity. These limitations underscore the need for more scalable and effective approaches for hierarchical clustering.

Given these findings, we take a critical perspective on recent research on deep hierarchical clustering and offer a straightforward yet effective alternative. Rather than designing specialized hierarchical clustering models, we de-

---

⋆Shared last authorship  [1]ETH AI Center, Zurich. [2]Department of Computer Science, ETH Zurich. Correspondence to: Emanuele Palumbo <emanuele.palumbo@inf.ethz.ch>.

*Proceedings of the $42^{nd}$ International Conference on Machine Learning*, Vancouver, Canada. PMLR 267, 2025. Copyright 2025 by the author(s).

velop a lightweight method to perform hierarchical clustering given a pre-trained flat model. In particular, we show that a lightweight algorithm implemented on top of (non-hierarchical) pre-trained models produces hierarchical clustering results that markedly outperform state-of-the-art dedicated models. Notably, our algorithm, which we name Logits to Hierarchies (L2H), *only uses logits* and *requires no fine-tuning* of the pre-trained model. Hence, it generally applies to black-box models even without access to internal representations (e.g., API calls to proprietary models) and bypasses the costly computation of pairwise distances between data points. Moreover, it also applies to supervised models, for which the inferred hierarchy of classes can aid model interpretability, e.g., for discovering potential biases such as spurious correlations between classes.

In summary, we make the following key contributions:

- We reveal significant limitations of recently proposed deep specialized methods for hierarchical clustering, highlighting their weaknesses on large-scale datasets and their subpar performance at the leaf level compared to non-hierarchical approaches.

- As an alternative approach, we propose a straightforward algorithm for hierarchical clustering that from the logits of a pre-trained flat model derives a hierarchical structure of clusters. Our method markedly outperforms specialized hierarchical models and has low computational requirements. With logits as its input, it computes a hierarchical clustering on ImageNet-sized datasets in under a minute on a CPU.

- To demonstrate how our method also applies to supervised models, we provide a case study on ImageNet, showing how the inferred hierarchy of classes recovers parts of the WordNet hierarchy, and helps to discover potential spurious correlations in the pre-trained model or biases in existing categorizations.

## 2. Related Work

Hierarchical clustering aims to learn clusters of data points that are organized in a hierarchical structure. The methods used can be broadly categorized into agglomerative and divisive approaches (Nielsen, 2016). The former tackle the problem with a bottom-up approach and iteratively agglomerate clusters into larger ones until a full hierarchy is built in the form of a dendrogram, starting with each datapoint being a separate cluster (Murtagh & Contreras, 2011). The similarity of data points is measured according to a distance function, which for high-dimensional data is often defined on a lower-dimensional representation space. Multiple linkage methods have been proposed to compute the distance between clusters of data points formed at a given step of the

algorithm (Sneath, 1957; Ward, 1963). As examples, *single*, *average*, and *complete* linkage characterize the distance between two clusters as the minimum, average, and maximum distance between their data points, respectively. Since these algorithms can be costly, particularly in high-dimensional spaces, approximate versions have been developed for faster computation (Abboud et al., 2019; Cochez & Mou, 2015). Notably, linkage methods are still widely applied in many domains as, for instance, in medical research (Nguyen et al., 2024; Senevirathna et al., 2023; Resende et al., 2023). Recent work also adopts them to assess how well the representations from pre-trained encoders generalize to cluster unseen datasets (Lowe et al., 2024).

On the other hand, divisive algorithms start with all objects belonging to the same cluster and recursively split them into subclusters. While early approaches are mostly based on heuristics, Dasgupta. (2016) proposed the Dasgupta cost: an objective function for evaluating a hierarchical clustering, with a divisive approach to provide an approximately optimal solution. HypHC introduces a continuous relaxation of Dasgupta's discrete optimization problem with provable guarantees via hyperbolic embeddings that better reflect the geometry of trees compared to Euclidean representations (Chami et al., 2020; Liu et al., 2019). More recently, research has focused on developing deep learning approaches for hierarchical clustering with specialized architectures (Mautz et al., 2020; Goyal et al., 2017; Shin et al., 2019; Vikram et al., 2019; Manduchi et al., 2023). Among these, DeepECT learns a hierarchical clustering on top of the embedding space of a jointly optimized autoencoder (Mautz et al., 2020). TreeVAE (Manduchi et al., 2023), on the other hand, learns a hierarchical clustering in the latent space of a variational autoencoder and provides a generative model that adheres to the learned hierarchy, thereby enabling sample generation in a structured manner (Manduchi et al., 2023). However, these approaches have mostly been tested on simple datasets, far from realistic settings. In our experiments in Section 4.1, we show that they present important limitations when deployed on more challenging datasets. We find these limitations to be linked to their high computational demands and their difficulty in modeling hierarchies that consist of a large number of leaf clusters.

Finally, the benefits of modeling a hierarchy in the data are not restricted to the unsupervised setup (Khrulkov et al., 2020; Linderman et al., 2023; Sinha et al., 2024). In particular, recent research focuses on leveraging a tree structure in the classes to assess and reduce the severity of misclassification of supervised models (Karthik et al., 2021). This can lead to safer models in cost-sensitive scenarios (Bertinetto et al., 2020) and allow a classifier to predict at different levels of the hierarchy depending on the required confidence (Goren et al., 2024). The visualization of hierarchies also provides global explanations of a model's functionality,

thereby improving a user's understanding of the model behavior and fostering trust (Chakraborty et al., 2017; Lipton, 2018).

## 3. Method

In this work, we take a critical perspective on a recent line of research on hierarchical clustering. After uncovering important limitations of recent hierarchical clustering approaches with deep specialized architectures, we propose an alternative strategy. Namely, rather than designing ad-hoc complex methods, we focus on adapting pre-trained flat models to output a hierarchy with minimal overhead. To this end, we introduce a lightweight algorithm to leverage the information contained in the logits of a pre-trained flat clustering model to output a hierarchy of clusters. In the following, we describe the proposed procedure and also provide a graphical illustration as well as detailed pseudocode.

Let $\mathcal{D} = \{\boldsymbol{x}_1, \ldots, \boldsymbol{x}_N\}$ be a dataset consisting of $N$ data points and $f_\theta$ be a non-hierarchical model trained to partition $\mathcal{D}$ into $K$ clusters. We assume that $f_\theta$ outputs unnormalized logits, from which the cluster assignment $k^*$ for a datapoint $\boldsymbol{x}$ is determined by taking the softmax

$$k^* = \underset{k \in \{1, \ldots, K\}}{\operatorname{argmax}} \operatorname{softmax}_k(f_\theta(\boldsymbol{x})) .$$

We define two functions

$$h_\theta(\boldsymbol{x}) = \underset{k \in \{1, \ldots, K\}}{\operatorname{argmax}} \operatorname{softmax}_k(f_\theta(\boldsymbol{x})) ,$$

$$g_\theta(\boldsymbol{x}) = \underset{k \in \{1, \ldots, K\}}{\max} \operatorname{softmax}_k(f_\theta(\boldsymbol{x})) ,$$

of which $h_\theta$ computes the cluster assignment for a datapoint $\boldsymbol{x}$, while $g_\theta$ computes the predicted probability of the cluster assignment for the datapoint $\boldsymbol{x}$.

A key idea behind our method is a simple yet effective way to determine the relatedness of clusters, or groups of clusters, while iteratively grouping them together to construct a hierarchy. Intuitively, to assess which group of clusters $G'$ is most related to a given group $G$, we propose the following strategy: for data points assigned to clusters in $G$, we determine which group $G'$ would have the majority of these data points reassigned to, if clusters in $G$ were not available.[1] Formally, we define the following functions to compute cluster assignments and corresponding predicted probabilities, restricting only to a subset of the total set of clusters.

We start with a masked version of the softmax function

$$\operatorname{m\_softmax}_k(\boldsymbol{v}; G) = \begin{cases} \frac{\exp(v_i)}{\sum_{j \in \{1, \ldots, K\} \setminus G} \exp(v_j)} & \text{if } i \notin G \\ 0 & \text{if } i \in G \end{cases}$$

given a $K$-dimensional vector $\boldsymbol{v}$ and a set $G \subset \{1, \ldots, K\}$. This function restricts the softmax operation to the elements of $\boldsymbol{v}$ at indexes in $\{1, \ldots, K\} \setminus G$. Next, we define

$$h_\theta^m(\boldsymbol{x}; G) = \underset{k \in \{1, \ldots, K\}}{\operatorname{argmax}} \operatorname{m\_softmax}_k(f_\theta(\boldsymbol{x}); G) ,$$

$$g_\theta^m(\boldsymbol{x}; G) = \underset{k \in \{1, \ldots, K\}}{\max} \operatorname{m\_softmax}_k(f_\theta(\boldsymbol{x}); G) ,$$

where the functions $h_\theta^m$ and $g_\theta^m$ correspond to $h_\theta$ and $g_\theta$ but restricting the choice of viable clusters to $\{1, \ldots, K\} \setminus G$. In particular, the function $h_\theta^m$ computes the cluster assignment for a datapoint $\boldsymbol{x}$ restricting to clusters in $\{1, \ldots, K\} \setminus G$, and $g_\theta^m$ computes the corresponding predicted probability. Lastly, we define

$$\mathcal{D}^c := \{\boldsymbol{x} \in \mathcal{D} \mid h_\theta(\boldsymbol{x}) = c\} ,$$

i.e., the subset of data points assigned to a given cluster $c \in \{1, \ldots, K\}$. Similarly, we denote as $\mathcal{D}^G = \cup_{c \in G} D^c$ the subset of data points assigned to a group of clusters $G \subset \{1, \ldots, K\}$.

---

**Algorithm 1** Logits to Hierarchies (L2H).

Given aggregation function $\Lambda$, and functions $g_\theta, g_\theta^m$ defined as above for pre-trained $K$-clustering model $f_\theta$.

---

**Input:** Dataset $\mathcal{D}$.
**Output:** Hierarchy H.
  # Groups initialized as single clusters
  Initialize groups $\mathcal{G} = \{G_1, \ldots, G_K\}$ where $G_k = \{k\}$
  # Hierarchy initialized as empty list
  H = []
  **for** step $t$ **from** 1 to $K - 1$ **do**
    **for** group $G$ **in** $\mathcal{G}$ **do**
      # Compute group scores as in Eqn. 1
      Compute $s(G) := \Lambda_{\boldsymbol{x} \in \mathcal{D}^c, c \in G} \, g_\theta(\boldsymbol{x})$
    **end for**
    # Select group with lowest score for merging
    Take $G^\star \in \operatorname{argmin}_{G \in \mathcal{G}} s(G)$
    **for** cluster $c$ **in** $\{1, \ldots, K\} \setminus G^*$ **do**
      # Compute total pred. probability per cluster as in Eqn.2
      Compute $rp(c) := \sum_{\substack{\boldsymbol{x} \in \mathcal{D}^{G^*} \\ h_\theta^m(\boldsymbol{x}; G^*) = c}} g_\theta^m(\boldsymbol{x}; G^*)$
    **end for**
    # Select $G^\dagger$, most related group to $G^*$, for merging
    Take $G^\dagger = \operatorname{argmax}_{G \in \mathcal{G} \setminus \{G^*\}} \frac{1}{|G|} \sum_{c \in G} rp(c)$
    # Update groups
    Update $\mathcal{G}$ by merging groups $G^*$ and $G^\dagger$
    # Update hierarchy
    Update H by adding that $G^*$ and $G^\dagger$ are merged at step $t$
  **end for**

---

We describe our proposed method in Algorithm 1. At the start of the procedure, $K$ groups are initialized as single clusters. [2] At each iteration, two groups are merged into a

---

[1]This passage is primarily for intuition and not strictly accurate. To be precise, we do not look at reassignments but rather at predicted probabilities of reassignments (see Equation (2)).

[2]Note that here *cluster* is used to refer to a single cluster found by the pre-trained model, while *group* refers to a set of clusters that are grouped together at a given step of the algorithm.

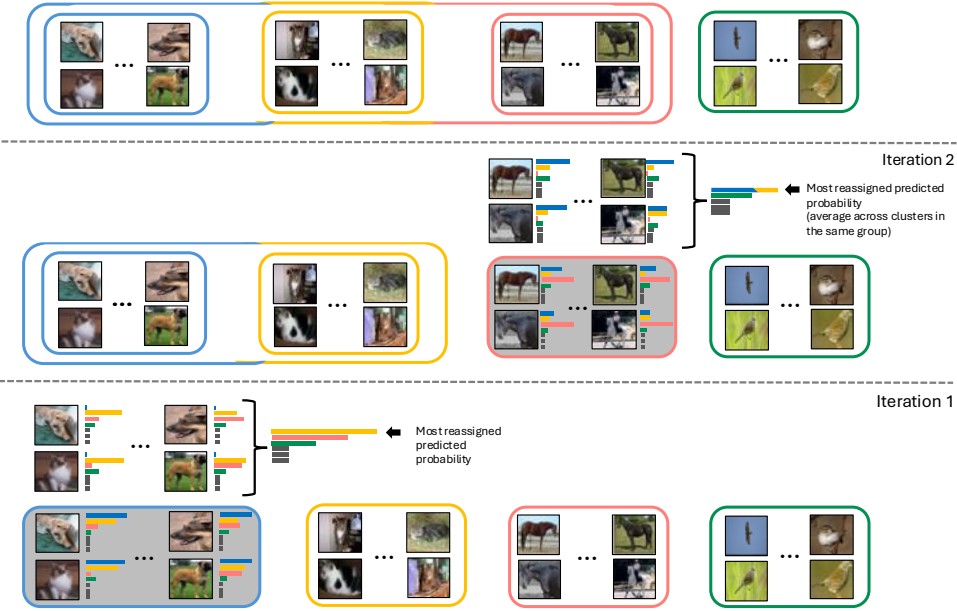

Figure 1: Illustration of the L2H algorithm. The four depicted clusters represent *dogs* in blue, *cats* in yellow, *horses* in red, *birds* in green respectively. In the first iteration (bottom), where groups correspond to single clusters, the dog cluster is selected for merging (shaded in grey). When recomputing predicted probabilities for samples in the dogs cluster, restricting to the remaining clusters, the cluster of cats has the highest predicted probability of reassignment. Note how, after merging, these two clusters are considered as a single group in the next iteration (top).

single group, constructing a tree of clusters up to the root in $K - 1$ iterations. Each iteration can be split into two stages. In the first stage, a score is computed for each group. To compute the score for a given group $G$ we aggregate the predicted probabilities for the data points assigned to clusters contained in $G$ as

$$s(G) = \bigwedge_{\substack{\boldsymbol{x} \in \mathcal{D}^c \\ c \in G}} g_\theta(\boldsymbol{x}) \tag{1}$$

where $\Lambda$ is a chosen aggregation function (e.g., the sum). Then, the lowest-scored group $G^*$ is selected for merging at this iteration, which concludes the first stage.

In the second stage, we search for the group $G^\dagger$ that is most related to $G^*$ to perform the merging. To do so, as mentioned above, we look at the subset of data points assigned to clusters in $G^*$: for these data points, we recompute cluster assignments and predicted probabilities, this time restricting to clusters not contained in $G^*$. More formally, the total reassigned predicted probability to each cluster not contained in $G^*$ is computed as

$$rp(c) := \sum_{\substack{\boldsymbol{x} \in \mathcal{D}^{G^*} \\ h_\theta^m(\boldsymbol{x}; G^*) = c}} g_\theta^m(\boldsymbol{x}; G^*) , \quad \forall c \in \{1, .., K\} \setminus G^* . \tag{2}$$

Note that this quantity can be interpreted as a measure of relatedness between each cluster $c \in \{1, .., K\} \setminus G^*$, and

the group of clusters $G^*$. The most related group to $G^*$ is finally selected as $G^\dagger \in \text{argmax}_{G \in \mathcal{G} \setminus \{G^*\}} \frac{1}{|G|} \sum_{c \in G} rp(c)$, i.e., by averaging the total reassigned predicted probability across clusters in each group and selecting the group with the highest average.

Given that cluster assignments and corresponding predicted probabilities can be computed via simple operations on the logits, the whole procedure can be executed given only the logits from a pre-trained model $f_\theta$ for the dataset $\mathcal{D}$. For further clarification, we provide a graphical illustration of the proposed grouping strategy (Figure 1) and an example Python implementation in Appendix A.

## 4. Experiments

In this section, we present the experimental results of our study. In the first part, we empirically demonstrate that existing specialized deep hierarchical clustering models face significant limitations in realistic scenarios. These limitations stem from their high computational demands and their difficulty in handling a large number of clusters. In contrast, our proposed method demonstrates compelling results on challenging vision datasets, achieving substantially better performance compared to these specialized models.

While the first part of this section focuses on hierarchi-

cal clustering, the second part extends our approach to supervised setups. We provide a case study showcasing the application of the L2H algorithm on top of a pre-trained ImageNet classifier. The results illustrate its potential for enhancing model interpretability and uncovering spurious correlations. Moreover, they complement the results for hierarchical clustering, demonstrating that our method can also be applied on top of pre-trained supervised models. Further details on datasets, implementations and metrics can be found in Appendix B.

## 4.1. Hierarchical Clustering

In this section, we evaluate the performance of recent specialized hierarchical clustering approaches on three challenging vision datasets— CIFAR-10, CIFAR-100 (Lake et al., 2015) and Food-101 (Bossard et al., 2014)— comparing their performance with our proposed method. We show our results in Table 1, including metrics to evaluate models at the leaf level and metrics to evaluate the quality of the produced hierarchy. To compare model performance at the leaf level, we report Normalized Mutual Information (NMI), Adjusted Rand Index (ARI), Accuracy and Leaf Purity (LP). To assess the quality of the hierarchical clustering, we report two metrics: Dendrogram Purity (DP) and Least Hierarchical Distance (LHD). The former was introduced in Kobren et al. (2017) and extends the notion of purity, normally evaluated at the leaf level, to assess the quality of a tree clustering: higher purity corresponds to higher quality of the hierarchy. Note that this metric was recently adopted in Manduchi et al. (2023) to benchmark deep hierarchical clustering models. Least Hierarchical Distance, on the other hand, measures the average minimal log-distance in the hierarchy between any pair of data points that have the same true label but different cluster assignments. A better hierarchy corresponds to a lower LHD. More details about our metrics can be found in Appendix B.3. For each dataset, we implement our L2H algorithm on top of two pre-trained flat clustering models, namely TURTLE (Gadetsky et al., 2024) and TEMI (Adaloglou et al., 2023). These are two recent clustering methods (see Appendix B.2 for more details), both of which are *not* designed to produce a hierarchy of clusters but only a flat clustering.

The results in Table 1 uncover the aforementioned limitations of recent deep hierarchical clustering methods (Deep-ECT, TreeVAE), which fail to achieve satisfactory performance even on moderately challenging datasets such as CIFAR-10. The results prove that these approaches struggle at modeling deep hierarchies, producing overly shallow trees in datasets with a large number of classes such as CIFAR-100 or Food-101. This is also linked to their high computational complexity. For instance, TreeVAE learns a hierarchical generative model with leaf-specific decoders: this choice helps its performance in a generative scenario

but impacts its scalability to large-scale datasets (see Table 2 for more details). Moreover, the comparison in terms of flat clustering metrics highlights that deep hierarchical models produce clusterings at the leaf level that are much less accurate than those obtained with non-hierarchical models.[3]

In contrast to alternative approaches, our proposed method recovers high-quality hierarchies for all three datasets when implemented with both TEMI or TURTLE as the backbone model. The results in the hierarchical metrics demonstrate that our method markedly outperforms existing approaches, with a consistent margin over costly deep learning specialized models. These findings demonstrate that the L2H algorithm can leverage the information embedded in the logits of a pre-trained flat clustering model to model an accurate hierarchy of the clusters. Most importantly, they show that our approach outperforms sophisticated deep hierarchical models while being more scalable and efficient. Note as well that our method does not require any fine-tuning of the pre-trained model, nor access to the internal representations. By construction, it retains the clustering performance of the pre-trained model at the leaf level, which matches state-of-the-art in our results. Importantly, the efficacy of our method is not hindered by the presence of a large number of classes in the dataset, as we witness for other methods. In particular, in Appendix C we show that our method achieves remarkable hierarchical clustering results on datasets as large as ImageNet-1k (Deng et al., 2009). As well we show in Appendix C that our approach is applicable across different choices for the backbone model, and notably, it outperforms deep specialized hierarchical approaches even when the chosen backbone model yields weaker clustering performance at the leaf level than TURTLE or TEMI. We also report additional ablations, and in particular we validate the robustness of our approach with respect to the hyperparameter $K$, corresponding to the number of clusters at the leaf level of the hierarchy.

In practice, hierarchical clustering results are often used as a visualization tool and to analyze the structure of a dataset at different levels of granularity. Hence, to evaluate our proposed approach, we visualize and inspect the hierarchy obtained with L2H-TURTLE on the CIFAR-100 dataset in Figure 2. Note that given the absence of leaf labels, we associate a class label to each leaf by looking at the most frequent label among the data points in the given leaf. While an off-the-shelf ground-truth hierarchy is not available for the CIFAR-100 dataset, the authors organize the 100 classes in 20 superclasses. Hence, we color-code the inferred leaf labels in the hierarchy by superclasses and check if the hierarchical clustering recovers this global structure. Notably, the global structure of the superclasses is largely

---

[3]In contrast, our approach retains the strong performance of the backbone model (TURTLE or TEMI) at the leaf level.

| | | Flat | | | | Hierarchical | | # leaves | Inference on test set |
|---|---|---|---|---|---|---|---|---|---|
| | | NMI (↑) | ARI (↑) | ACC (↑) | LP (↑) | DP (↑) | LHD (↓) | | |
| **CIFAR-10** | Agglomerative | 0.074 | 0.038 | 0.211 | 0.246 | 0.121 | 0.549 | 10 | ✗ |
| | HypHC | 0.019 | 0.009 | 0.134 | 0.359 | 0.104 | 0.569 | 10 | ✗ |
| | DeepECT | 0.006 | 0.002 | 0.110 | 0.110 | 0.101 | 0.369 | 2-3 | ✓ |
| | TreeVAE | 0.414 | 0.313 | 0.497 | 0.523 | 0.341 | 0.410 | 10 | ✓ |
| | L2H-TEMI | 0.907 | 0.906 | 0.956 | 0.957 | 0.902 | 0.348 | 10 | ✓ |
| | L2H-Turtle | **0.985** | **0.989** | **0.995** | **0.995** | **0.988** | **0.277** | 10 | ✓ |
| **CIFAR-100** | Agglomerative | 0.223 | 0.020 | 0.090 | 0.131 | 0.019 | 0.428 | 100 | ✗ |
| | HypHC | 0.072 | 0.004 | 0.031 | 0.560 | 0.011 | 0.499 | 100 | ✗ |
| | DeepECT | 0.016 | 0.005 | 0.070 | 0.070 | 0.052 | 0.121 | 2-3 | ✓ |
| | TreeVAE | 0.199 | 0.098 | 0.228 | 0.242 | 0.103 | 0.484 | 20 | ✓ |
| | L2H-TEMI | 0.778 | 0.565 | 0.682 | 0.701 | 0.502 | 0.298 | 100 | ✓ |
| | L2H-Turtle | **0.917** | **0.831** | **0.896** | **0.897** | **0.803** | **0.235** | 100 | ✓ |
| **Food-101** | Agglomerative | 0.082 | 0.004 | 0.039 | 0.045 | 0.011 | 0.438 | 101 | ✗ |
| | HypHC | 0.035 | 0.002 | 0.022 | 0.630 | 0.011 | 0.573 | 101 | ✗ |
| | DeepECT | 0.003 | 0.000 | 0.011 | 0.011 | 0.010 | 0.333 | 2-3 | ✓ |
| | TreeVAE | 0.114 | 0.017 | 0.057 | 0.058 | 0.016 | 0.483 | 20 | ✓ |
| | L2H-TEMI | **0.917** | **0.841** | **0.904** | **0.913** | **0.801** | **0.270** | 101 | ✓ |
| | L2H-Turtle | 0.894 | 0.800 | 0.876 | 0.877 | 0.758 | 0.297 | 101 | ✓ |

Table 1: Quantitative comparison of hierarchical clustering performance on three datasets (CIFAR-10, CIFAR-100, Food-101). We report as a baseline agglomerative clustering, deep hierarchical specialized models (DeepECT, TreeVAE), and our L2H method applied on top of two state-of-the-art flat models (TEMI, TURTLE). We also indicate the number of leaves in the hiearchy modelled by each approach, and whether a given method can perform inference on a hold-out test set. We bold best results for each metric and underline results that are artifacts of degenerate solutions with shallow hierarchies. Notably the application of L2H does not affect flat clustering performance, retaining the clustering performance of the pre-trained model (TURTLE, TEMI) at the leaf level.

| | **Dataset** | | | |
|---|---|---|---|---|
| | **CIFAR-10** $K = 10$ $N_{tr} = 50000$ | **CIFAR-100** $K = 100$ $N_{tr} = 50000$ | **Food-101** $K = 101$ $N_{tr} = 75750$ | **ImageNet1K** $K = 1000$ $N_{tr} = 1281167$ |
| L2H | $< 0.01$ | $< 0.01$ | $< 0.01$ | $0.45 \pm 0.0$ |
| Agglomerative | $< 0.1$ | $< 0.1$ | $0.8$ | - |
| HypHC | $163.7 \pm 4.0$ | $153.3 \pm 19.4$ | $195.3 \pm 3.2$ | - |
| DeepECT | $24.1 \pm 15.1$ | $26.2 \pm 9.8$ | $67.5 \pm 36.6$ | - |
| TreeVAE | $364.1 \pm 76.8$ | $756.3 \pm 178.6$ | $2293.7 \pm 211.7$ | - |
| L2H-TURTLE | $1.6 \pm 0.0$ | $1.6 \pm 0.0$ | $1.7 \pm 0.0$ | $5.25 \pm 0.0$ |

Table 2: Training time (in minutes) for our proposed method compared to baselines for hierarchical clustering on CIFAR-10, CIFAR-100, and Food-101 datasets. At the top, we report the runtime for the L2H algorithm alone. Below, we report the runtime of the TURTLE model plus our L2H algorithm to produce a hierarchy, compared with the runtime of each baseline model. Results are averaged over three runs and include standard deviations.

reflected in the visualized hierarchy. Most interesting is that the outliers, for which the color does not coincide with the neighboring leaves, still reflect meaningful semantic associations. For instance, *whale* and *dolphin*—despite being aquatic mammals—are grouped with fish species. However, this is not surprising, given their adaptation exclusively to aquatic environments and the presence of similar traits to fishes, like streamlined bodies. On the contrary, mammals such as *otter*, *beaver*, and *seal*, which are only semi-aquatic, are grouped with other small to medium-sized terrestrial mammals, emphasizing size and communal characteristics like the presence of limbs and fur. Another example is the characterization of *worm* and *snake* alongside in the hierarchy. Although snakes are reptiles, their elongated, limbless bodies visually resemble those of non-insect invertebrates like worms. This showcased analysis confirms the efficacy of our method in recovering a tree structure that follows meaningful semantic associations. The results indicate that

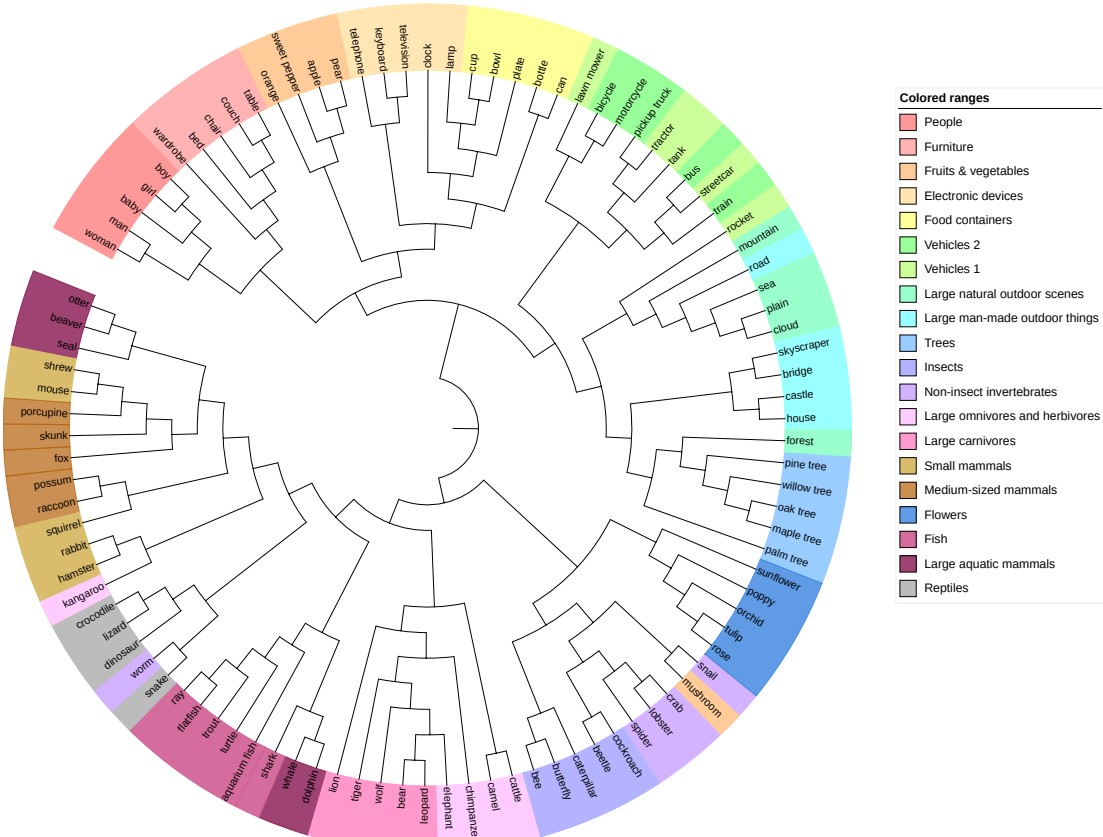

Figure 2: Visualization of the hierarchical clustering produced by L2H-TURTLE on the CIFAR-100 dataset. The inferred hierarchy is represented as a circular tree. On the lowest level, the leaves are annotated by reporting the most frequent label for the samples in each leaf. Leaves are color-coded according to the 20 superclasses in the dataset.

our method produces hierarchies that enable detailed exploration of the structure in the data at varying levels of granularity. Inspecting the hierarchy gives valuable insights for interpretability, revealing underlying associations by the model.

We end this section with a comparison in terms of the computational cost of our method compared with alternative models, and in particular with specialized deep learning approaches for hierarchical clustering. As the results in Table 2 show, our proposed L2H algorithm is extremely lightweight. To compare with the efficiency of alternative methods, we also measure the overall runtime to perform hierarchical clustering with L2H-TURTLE, which includes the runtime to train the TURTLE model, as an example of backbone model. Our approach allows us to perform hierarchical clustering extremely efficiently, even on large-scale datasets such as ImageNet-1k, with a total training time of a few minutes. Note that, due to the combined efficiency of our method and state-of-the-art flat clustering models like TURTLE, the overall runtime scales seamlessly with dataset size and number of leaves in the hierarchy. Conversely,

deep hierarchical approaches exhibit a significantly higher computational cost. Moreover, as with TreeVAE, increasing dataset size and number of classes markedly increases the computational burden. Finally, it is to note that TURTLE, as well as TEMI, leverages CLIP embeddings for training. Hence, to further validate the superior efficiency and efficacy of our method compared to alternative deep specialized approaches, in Appendix C we compare our results with the ones obtained by training TreeVAE using CLIP embeddings. These results again validate that our L2H approach achieves markedly superior performance when compared models have access to foundation model embeddings for training. Once again, this gain in performance comes also with much higher computational efficiency.

### 4.2. Case Study: Pre-trained ImageNet Classifier

In this section we complement the results from the previous section by showing that our method can be applied in a supervised setup, producing a hierarchy given the logits of a pre-trained classifier. Specifically, we use the ImageNet-1k dataset (Deng et al., 2009), which comprises over a million

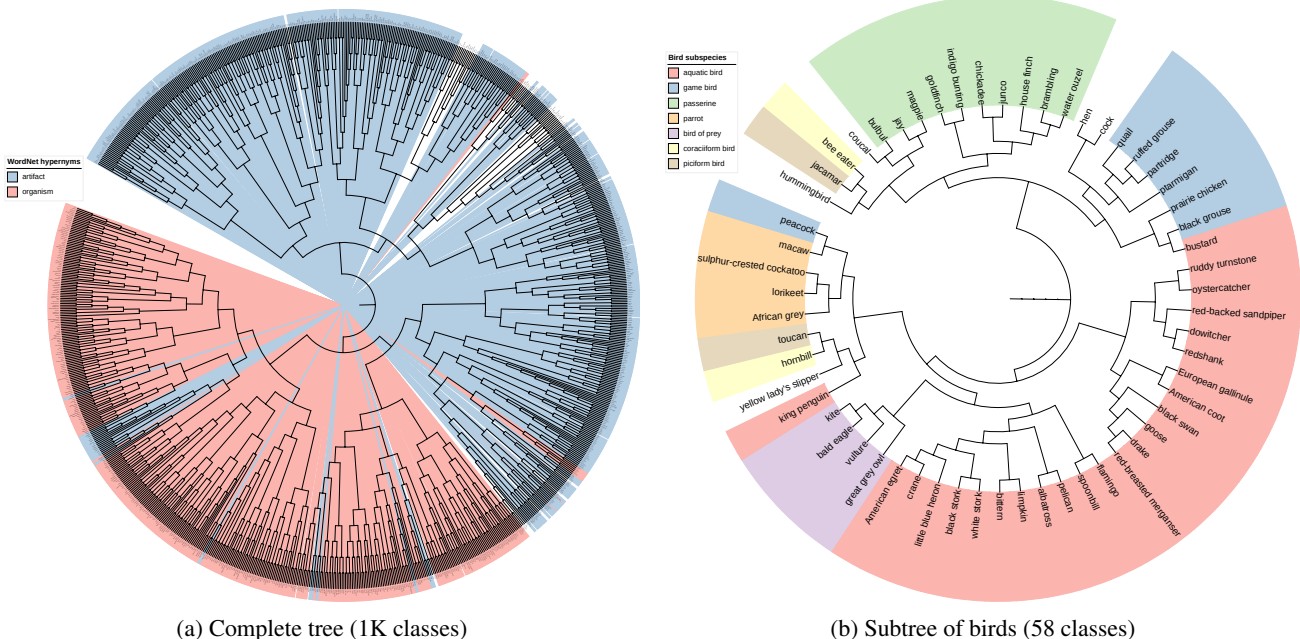

(a) Complete tree (1K classes)

(b) Subtree of birds (58 classes)

Figure 3: Visualization of inferred hierarchy for the ImageNet-1K dataset. The hierarchy is represented as a circular tree, where the leaf nodes are organized in a circle. Figure 3a shows the complete tree colored by the corresponding WordNet hypernyms "artifact" and "organism", which are the largest two superclasses in the ImageNet dataset. Figure 3b shows the subtree of birds colored by different bird species if they comprise more than one class. The results show that our method recovers a significant portion of the global and local hierarchical structure of the ImageNet dataset.

images and a thousand distinct classes, with an underlying hierarchical structure organized according to the WordNet hierarchy. We apply the L2H algorithm on the logits of a pre-trained ImageNet classifier to model the hierarchy of classes. As a pre-trained classifier, we use the InternImage model (Wang et al., 2023).

Figure 3a illustrates the inferred hierarchy for the thousand ImageNet classes. The colors indicate whether a leaf node corresponds to the superclass "artifact" or "organism", which are the largest two superclasses that can be determined based on the corresponding WordNet hypernyms of each class in ImageNet. Overall, we observe a distinct separation between the two superclasses in the inferred tree.

In addition, Figure 3b shows a subtree of the inferred hierarchy that comprises different bird species. Specifically, it shows 58 of the 60 classes of birds contained in the ImageNet dataset. The leaf nodes are colored by different clades of bird species (based on the WordNet hierarchy), showing that the inferred hierarchy groups together related species. For example, the group "aquatic bird" is almost completely represented in one of the two main branches, which further splits into a separate cluster for "parrots" and another one for "bird of prey". The other main branch of the tree subdivides further into "passerine" and "game bird" forming distinct clusters.

Overall, our results suggest that the inferred hierarchy recovers a significant portion of the global and local hierarchical structure of the ImageNet dataset given the logits of a pre-trained ImageNet classifier trained with non-hierarchical labels. Yet, the inferred tree also reveals interesting outliers. For example, in Figure 3a, there is a distinct subtree for snow-related artifacts (e.g., dogsled, snowmobile, bobsled) within a large branch of the tree that comprises organisms. Further investigation shows that this group of artifacts is merged with arctic animals (e.g., malamute, Siberian husky, Eskimo dog), which reveals a spurious correlation between classes and highlights potential biases of the pre-trained model. Likewise, in Figure 3b, we see potential outliers such as "bustard" among game birds. Interestingly, this constitutes an example where our method reveals an ambiguity in the WordNet hierarchy, which classifies "bustard" as a wading bird, while it is usually defined as a terrestrial game bird, like in our inferred categorization.

## 5. Conclusion

In this work, we uncover significant limitations of existing deep hierarchical clustering models, demonstrating that these methods face important challenges in scaling to large-scale datasets, falling short of delivering satisfactory performance. As an alternative approach, we propose

a lightweight yet effective procedure for hierarchical clustering based on pre-trained non-hierarchical models. Notably, our solution proves to be markedly more effective and significantly more computationally efficient than existing methods. Specifically, we demonstrate that our method can successfully handle large datasets with hundreds of classes, taking an important step for the practical applicability of hierarchical clustering methods in realistic settings. Moreover, we show that the usefulness of our approach extends to supervised setups by implementing it on top of a pre-trained classifier to recover a meaningful hierarchy of classes. A case study on ImageNet shows that our approach provides relevant insights for interpretability, and can reveal potential biases in the pre-trained model or spurious correlation in the data.

Our proposed method is general and may be applied to different data types beyond vision, which we leave as an opportunity for future work. Another direction for future work is the investigation of strategies for automatically selecting meaningful levels of the inferred hierarchy. Hierarchical clustering presents important advantages over non-hierarchical clustering by simultaneously capturing the structure in the data at multiple levels of granularity. However, manually inspecting the hierarchy is still necessary to extract valuable insights. Thus, future work could investigate strategies to partly bypass this process, automatically selecting levels of the hierarchy that provide the most meaningful clustering. Our proposed method relies on the assumption that logits from a pre-trained flat model can be used as a proxy to measure cluster similarities. While empirically supported by our experimental results, this assumption may not always hold, for instance if the pre-trained model is poorly calibrated.

## Acknowledgements

EP is supported by a fellowship from the ETH AI Center, and received funding from the grant #2021-911 of the Strategic Focal Area "Personalized Health and Related Technologies (PHRT)" of the ETH Domain (Swiss Federal Institutes of Technology). AR is supported by the StimuLoop grant #1-007811-002 and the Vontobel Foundation. The authors are grateful to Laura Manduchi for useful discussions.

## Impact Statement

As other ML models, clustering methods—such as the ones explored in this work—are vulnerable to the risk of reflecting potential biases and spurious correlations from the data they are trained on. However, hierarchical approaches offer a promising direction for mitigating these risks by enhancing the interpretability and transparency of clustering and classification outcomes.

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

# Appendix

## A. Code Implementation of the L2H Algorithm

In Figure 4, we provide a Python implementation of the L2H algorithm proposed in this work using standard scientific computing libraries (NumPy, SciPy). As stated in Section 3, our algorithm only requires the logits as input. It can be executed on the CPU even for large datasets, e.g., with a runtime of less than a minute for ImageNet. Note that our procedure can be applied to any pre-trained unsupervised model to perform hierarchical clustering. Further, it can also be applied to logits from a supervised model to infer a hierarchy of classes. We store the hierarchy as a list comprised of groups of clusters that are merged iteratively. The aggregation function for computing the score per group is a design choice (as described in Appendix B.2) that can be viewed as a hyperparameter.

```python
import numpy as np
from scipy.special import softmax

def L2H(logits):
    """
    L2H Algorithm.
    Args:
        logits: Logits from model (N x K) where N number of datapoints  in the dataset
                and K is the number of clusters
    Returns:
        steps: Merging steps characterizing the hierarchy

    """
    # Number of clusters is equal to size of last dimension in the logits
    K = logits.shape[-1]
    # Initialize groups of clusters to single clusters
    groups = [(c,) for c in range(K)]
    # Initialize list of steps that characterize hierarchy
    steps = []
    # Given the logits for the whole dataset, compute assignments and predicted probabilities
    softmaxed_logits = softmax(logits, axis=-1)
    assignments = np.argmax(softmaxed_logits, axis=-1)
    pred_probs = np.max(softmaxed_logits, axis=-1)
    for step in range(1, K):
        # Compute score for for each group (which chosen aggregation function)
        score_per_gr = {}
        for group in groups:
            score_per_gr[group] = sum([np.mean(pred_probs[assignments == c]) for c in group])
        # Get the group with the lowest score (lsg), will be merged at this iteration
        lsg = min(score_per_gr, key=score_per_gr.get)
        # Get the logits for datapoints assigned to the lowest score group
        logits_lsg = logits[np.where(np.isin(assignments, lsg))[0]]
        # Reassign datapoints in lsg to only clusters not in lsg,
        # and re-compute predicted probabilities
        msm_logits_lsg = np.zeros_like(logits_lsg)
        cls_not_in_lsg = [c for c in range(K) if c not in lsg]
        cls_in_lsg = [c for c in range(K) if c in lsg]
        msm_logits_lsg[:, cls_not_in_lsg] = softmax(logits_lsg[:, cls_not_in_lsg], axis=-1)
        msm_logits_lsg[:, cls_in_lsg] = 0.
        reassignments = np.argmax(msm_logits_lsg, axis=-1)
        re_pred_probs = np.max(msm_logits_lsg, axis=-1)
        # Compute the total reassigned predicted probability per cluster and average across
        # clusters in each group.Then select the group with the highest average.
        re_pp_per_group = {
            group: np.mean([np.sum(re_pred_probs[reassignments == c]) for c in group]) for
            group in groups if group != lsg
        }
        mtg = max(re_pp_per_group, key=re_pp_per_group.get)
        # Merge `lsg` with `mtg` and update `groups`.
        groups = [gr for gr in groups if gr not in [lsg, mtg]] + [lsg + mtg]
        # Add merging in current iteration to steps
        steps.append((lsg, mtg))
    return steps
```

Figure 4: Python code implementation for the L2H algorithm presented in Section 3. Note that we choose the aggregation function when computing the score per group as described in Appendix B.2.

## B. Experimental Details

### B.1. Datasets

In this work, we run experiments on five challenging vision datasets, namely CIFAR-10 and CIFAR-100 (Lake et al., 2015), Food-101 (Bossard et al., 2014), ImageNet1K (Deng et al., 2009), and as well INaturalist21(Van Horn et al., 2021) introduced in Appendix C. CIFAR-10 and CIFAR-100 are well-established object classification datasets. The CIFAR-10 dataset consists of 60000 32x32 colored images, divided in 10 classes: *airplane*, *automobile*, *bird*, *cat*, *deer*, *dog*, *frog*, *horse*, *ship*, *truck*. The train/test splits contain 50000 and 10000 images respectively. Similarly, also the CIFAR-100 dataset consists of 60000 32x32 colored images. However, they are organized into 100 classes. In addition, the 100 classes are grouped into 20 superclasses. As for CIFAR-10, the train/test splits also contain 50000 and 10000 images respectively. The Food101 dataset is a fine-grained classification dataset of food images, consisting of 101000 images for 101 classes. Images are high-resolution, up to 512 pixels side length. Images are split between 75750 training samples and 25250 test images. The ImageNet-1k dataset, widely used in computer vision, consists of 1000 classes organized according to the WordNet hierarchy (Miller, 1995), with 1281167 training and 50000 test samples, respectively. The INaturalist21 dataset (Van Horn et al., 2021) contains 2.7 million images of natural species labelled at different taxonomy levels.

### B.2. Implementation Details

For our hierarchical clustering experiments, to train the TURTLE and TEMI models on all considered datasets we use the official code provided by the authors with recommended choices for hyperparameters (Gadetsky et al., 2024; Adaloglou et al., 2023). In particular, TEMI employs CLIPViTL/14 representations of the data, while TURTLE employs both CLIPViTL/14 and DINOv2 ViT-g/14 representations. For more details on TURTLE trained using two representation spaces, see the original paper (Gadetsky et al., 2024). We train both TEMI and TURTLE with a number of clusters $K$ equal to the true number of classes in each dataset (unless explicitly stated otherwise, e.g. in the sensitivity analysis reported in Figure 5). For each dataset, we train models on the training set, then report metrics on the test set. Note that the L2H algorithm takes as input logits from the training set to infer the hierarchy, while metrics that evaluate the quality of the hierarchy are computed on the test set. As the aggregation function $\Lambda$ in the L2H algorithm (see Section 3) we employ

$$\bigwedge_{\substack{\boldsymbol{x} \in \mathcal{D}^c \\ c \in G}} g_\theta(\boldsymbol{x}) = \sum_{c \in G} \frac{1}{|\mathcal{D}^c|} \sum_{\boldsymbol{x} \in \mathcal{D}^c} g_\theta(\boldsymbol{x})$$

which we find to work well experimentally. However, other choices are possible (see also Table 7). We implement TreeVAE (Manduchi et al., 2023) with their contrastive approach using the provided PyTorch codebase with corresponding defaults. The splitting criterion is set to the number of samples, an inductive bias that benefits this baseline method, since all datasets are balanced (Manduchi et al., 2023; Vandenhirtz et al., 2024). We set the number of clusters to 10 for CIFAR-10 and to 20 for the rest, due to the computational complexity, as seen in Table 2, as well as memory complexity, since every additional leaf adds a new decoder. DeepECT (Mautz et al., 2020) is also implemented using their provided codebase with the augmented version. Note that similar to the results shown in Manduchi et al. (2023), for colored datasets, DeepECT fails to grow trees, as they always collapse, indicating that DeepECT fails to find meaningful splits. We implement agglomerative clustering using the scikit-learn library (Pedregosa et al., 2011), and fit the model using PCA embeddings of the datasets with 50 components and Ward's criterion (Ward, 1963) as the linkage method. Using the author's original codebase, we further train Hyperbolic Hierarchical Clustering (Liu et al., 2019) on CLIP embeddings of the respective datasets. The authors do not describe how to retrieve cluster assignments using their method, so we follow the agglomerative clustering procedure and assume the leaves of the last $K$ tree nodes created to form a cluster, where $K$ corresponds to the chosen number of clusters.

### B.3. Metrics

Here we provide more details on the metrics reported in our experiments in Section 4.1. In our comparisons, we evaluate models both on flat and hierarchical clustering.

**Flat clustering** To assess model performance in flat clustering, for each model we take the clustering at the level of the hierarchy where the number of clusters corresponds to the true number of classes $K$ in a given dataset. If the number of leaves at the leaf level of the hierarchy is smaller than $K$, as is the case for, e.g., TreeVAE and DeepECT on CIFAR-100, we consider the clustering at the leaf level. For flat clustering comparisons, we resort to well-established metrics, namely

NMI, ARI, Accuracy, and Purity of the clusters (i.e., Leaf Purity). To compute accuracy and leaf purity, we resort to recent implementations in (Gadetsky et al., 2024) and (Manduchi et al., 2023), respectively.

**Hierarchical clustering**  To assess the quality of a learned hierarchy, and compare the results of different models in hierarchical clustering, we resort to two metrics. Dendrogram Purity (DP), introduced in (Kobren et al., 2017), extends the notion of leaf purity to evaluate the purity of hierarchical clusters, and was recently adopted to benchmark hierarchical clustering models (Manduchi et al., 2023). Following the notation of (Kobren et al., 2017), let $\mathcal{C}^*$ denote the true $K$-clustering (i.e., true class labeling) of a dataset $\mathcal{D}$. Then define

$$\mathcal{P}^* = \Big\{(x_i, x_j) \forall x_i, x_j \in \mathcal{D}, x_i \neq x_j \mid C^*(x_i) = C^*(x_j)\Big\}$$

as the set of pairs of data points that belong to the same true cluster. Dendrogram Purity (DP) is then defined for a hierchical clustering $\mathcal{H}$ as

$$DP(\mathcal{H}) = \frac{1}{|\mathcal{P}^*|} \sum_{k=1}^{K} \sum_{(x_i, x_j) \in \mathcal{C}_k^*} \mathrm{pur}(\mathrm{lvs}(\mathrm{LCA}(x_i, x_j)), \mathcal{C}_k^*),$$

where $\mathrm{LCA}(x_1, x_2)$ computes the least common ancestor node of data points $x_1$ and $x_2$ in $\mathcal{H}$, $\mathrm{lvs}(z)$ returns the set of leaves of the sub-tree rooted at any internal node $z$, $\mathrm{pur}(S_1, S_2) = |S_1 \cap S_2|/|S_1|$, and $\mathcal{C}_k^*$ is the set of data points belonging to the true cluster $k$. One possible caveat of this metric is its high correlation with Leaf Purity: with a high leaf purity, most pairs of samples sharing the true label will inevitably fall into the same leaf. To address this, we introduce an additional metric for evaluation, namely Least Hierarchical Distance. With a similar notation as above we define

$$\bar{\mathcal{P}}^* = \Big\{(x_i, x_j) \forall x_i, x_j \in \mathcal{D}, x_i \neq x_j \mid C^*(x_i) = C^*(x_j) \wedge l(x_i; \mathcal{H}) \neq l(x_j; \mathcal{H})\Big\}$$

where the function $l(x; \mathcal{H})$ returns the cluster prediction for datapoint $x$ at the leaf level of $\mathcal{H}$. Hence $\bar{\mathcal{P}}^*$ is the set of all pairs of points sharing the same true label that are *not* assigned to the same leaf in $\mathcal{H}$. Least Hierarchical Distance is then defined for a hierarchical clustering $\mathcal{H}$ as

$$LHD(\mathcal{H}) = \frac{1}{|\bar{\mathcal{P}}^*|} \sum_{(x_i, x_j) \in \bar{\mathcal{P}}^*} \frac{\log_2(td(l(x_i; \mathcal{H}), l(x_j; \mathcal{H}))) - 1}{\log_2(K) - 1}$$

where $td(l_1, l_2)$ computes the number of edges in the shortest path that connects two leaves $l_1, l_2$ in the tree defined by $\mathcal{H}$. Different from Dendrogram Purity, Least Hierarchical Distance only takes into consideration pairs of data points with the same true label that do not fall into the same leaf. Hence, it does not exhibit a strong correlation with Leaf Purity, being more specific to the quality of the hierarchy rather than influenced by the clustering at the leaf level.

## C. Additional Results and Visualizations

Table 3 presents the results of our L2H algorithm implemented on top of the TURTLE backbone model for hierarchical clustering on the ImageNet1K dataset (Deng et al., 2009). These results complement the ones shown in Table 1, demonstrating that our method achieves outstanding performance for hierarchical clustering, even on datasets with large scale and numerous classes. It is worth noting that a direct comparison with alternative approaches (e.g., DeepECT, TreeVAE) is not feasible in this setting, as these methods lack the scalability to handle datasets of this magnitude and complexity. The quantitative results underscore the effectiveness of our method, successfully uncovering a meaningful hierarchical structure in a dataset of this size and complexity.

|  | NMI ($\uparrow$) | ARI ($\uparrow$) | ACC ($\uparrow$) | LP ($\uparrow$) | DP ($\uparrow$) | LHD ($\downarrow$) |
|---|---|---|---|---|---|---|
| L2H-TURTLE | 0.882 | 0.621 | 0.726 | 0.744 | 0.560 | 0.210 |

Table 3: Hierarchical clustering performance of our L2H method applied on top of the TURTLE pre-trained model on the ImageNet1K dataset.

In Table 4, we present results obtained by applying our proposed approach using an alternative backbone model to TEMI and TURTLE, namely the TCL flat clustering model (Yunfan et al., 2022). Unlike TEMI and TURTLE, TCL does not

rely on pre-trained foundation model representations. This ablation highlights the generality of our method, demonstrating that it can be effectively applied with diverse choices of the backbone model. Notably, our approach maintains good hierarchical clustering performance even when the underlying backbone exhibits weaker flat clustering capabilities than TEMI or TURTLE, still outperforming deep specialized hierarchical approaches (see Table 1).

|  |  | Flat | | | | Hierarchical | |
|---|---|---|---|---|---|---|---|
|  |  | NMI (↑) | ARI (↑) | ACC (↑) | LP (↑) | DP (↑) | LHD (↓) |
| **CIFAR-10** | L2H-TCL | 0.785 | 0.744 | 0.868 | 0.877 | 0.733 | 0.398 |
| **CIFAR-100** | L2H-TCL | 0.547 | 0.215 | 0.343 | 0.437 | 0.218 | 0.351 |
| **Food-101** | L2H-TCL | 0.455 | 0.168 | 0.279 | 0.348 | 0117 | 0.396 |

Table 4: Results obtained applying our L2H method with the TCL model used as backbone on CIFAR-10, CIFAR-100, Food-101 datasets.

In this work, we apply our L2H algorithm on top of flat models (TURTLE and TEMI) that both leverage CLIP embeddings to cluster the data. Hence, for further comparisons in Table 5 we report the results obtained with agglomerative clustering with Ward's linkage performed on pre-trained CLIP embeddings from CIFAR-10, CIFAR-100, and Food-101 datasets. While leveraging pre-trained CLIP embeddings improves the performance of agglomerative clustering, a direct comparison with the performance of L2H-TEMI (see Table 1), which is also solely based on CLIP embeddings, shows that our method still markedly outperforms this baseline.

|  |  | Flat | | | | Hierarchical | |
|---|---|---|---|---|---|---|---|
|  |  | NMI (↑) | ARI (↑) | ACC (↑) | LP (↑) | DP (↑) | LHD (↓) |
| **CIFAR-10** | Agg. (CLIP embeddings) | 0.799 | 0.724 | 0.805 | 0.826 | 0.716 | 0.442 |
| **CIFAR-100** | Agg. (CLIP embeddings) | 0.690 | 0.386 | 0.531 | 0.637 | 0.341 | 0.343 |
| **Food-101** | Agg. (CLIP embeddings) | 0.868 | 0.730 | 0.837 | 0.872 | 0.703 | 0.299 |

Table 5: Quantitative results for flat and hierarchical clustering performance of agglomerative clustering performed on CLIP embeddings.

To provide further comparisons that complement the results in Section 4.1, Table 6 presents results on the CIFAR-100 dataset comparing our proposed approach alongside two additional methods. First, we evaluate TreeVAE (Manduchi et al., 2023) trained on CLIP embeddings of CIFAR-100, demonstrating that our method achieves substantially better performance. We include this comparison as both TEMI and TURTLE have access to CLIP embeddings for training. Hence this comparison shows that, even in a setup where deep specialized hierarchical approaches share access to foundation model embeddings, our proposed approach achieves significantly better performance with much higher efficiency. In particular, training TreeVAE in this setup requires hours on a GPU, while training L2H-TURTLE requires under two minutes. Furthermore, we include a comparison with the recent method introduced by Lowe et al. (2024), which combines UMAP dimensionality reduction with Ward's agglomerative clustering for zero-shot clustering with pre-trained embeddings. Since this approach relies on agglomerative clustering, it presents the drawback of not allowing inference on a separate test set. Hence, we both train and evaluate the this method on the test set, which results in an unfair advantage. Despite this, our method still significantly outperforms this additional baseline.

|  | NMI (↑) | ARI (↑) | ACC (↑) | LP (↑) | DP (↑) | LHD (↓) |
|---|---|---|---|---|---|---|
| TreeVAE+CLIP | 0.665 | 0.285 | 0.255 | 0.255 | 0.181 | 0.285 |
| Lowe et al (2024) | 0.753 | 0.499 | 0.616 | 0.664 | 0.445 | 0.303 |
| L2H-TEMI | 0.778 | 0.565 | 0.682 | 0.701 | 0.502 | 0.298 |
| L2H-TURTLE | 0.917 | 0.831 | 0.896 | 0.897 | 0.803 | 0.235 |

Table 6: Performance comparison on the CIFAR-100 dataset of our L2H approach, using TURTLE and TEMI as backbone models, with the method from Lowe et al (2024) and TreeVAE trained on CLIP embeddings. Hierarchical and flat clustering metrics are reported.

In Table 7, we provide an ablation that reports the results of L2H-TURTLE on the hierarchical clustering experiments from Section 4.1 with different choices for the aggregation function $\Lambda$ in the L2H algorithm. The results indicate that tweaks in the aggregation function alter performance, though without abrupt changes in the metrics. These results also motivate our designated choice of aggregation function—corresponding to the last row—which works well experimentally.

| | $\Lambda$ | CIFAR-10 | | CIFAR-100 | | Food-101 | |
|---|---|---|---|---|---|---|---|
| | | DP ($\uparrow$) | LHD ($\downarrow$) | DP ($\uparrow$) | LHD ($\downarrow$) | DP ($\uparrow$) | LHD ($\downarrow$) |
| | $\sum_{c \in G} \sum_{\boldsymbol{x} \in \mathcal{D}^c} g_\theta(\boldsymbol{x})$ | 0.988 | 0.258 | 0.801 | 0.244 | 0.758 | 0.294 |
| L2H-TURTLE | $\frac{1}{|G|} \sum_{c \in G} \frac{1}{|\mathcal{D}^c|} \sum_{\boldsymbol{x} \in \mathcal{D}^c} g_\theta(\boldsymbol{x})$ | 0.988 | 0.248 | 0.793 | 0.283 | 0.751 | 0.335 |
| | $\sum_{c \in G} \frac{1}{|\mathcal{D}^c|} \sum_{\boldsymbol{x} \in \mathcal{D}^c} g_\theta(\boldsymbol{x})$ | 0.988 | 0.277 | 0.803 | 0.235 | 0.758 | 0.297 |

Table 7: Results for hierarchical clustering, in terms of Dendrogram Purity and Least Hierarchical Distance, implementing the L2H algorithm with different choices for the aggregation function $\Lambda$, on top of the TURTLE model.

In Figure 5 we provide an ablation to test the sensitivity of our approach to the value of the hyperparameter $K$, corresponding to the number of leaves in the hierarchy and the number of clusters set for the pre-trained flat model. In particular, we report the results for L2H-TURTLE on the CIFAR-100 dataset varying the value of $K$ across the range $\{85, 90, 95, 100, 105, 110, 115\}$. Note that this range is symmetric around the true number of classes/clusters equal to 100. Therefore, we both explore the case of over- and under-estimating the true number of clusters at the leaf level of the hierarchy. We report the results for both flat (NMI, ARI, ACC, LP) and hierarchical (DP, LHD) metrics. Best performance across all metrics is achieved when $K$ is set to the true number of clusters, while the performance gracefully degrades when $K$ is set to be an over- or under-estimated value. This demonstrates the robustness and stability of our approach with respect to this hyperparameter, which is particularly important in practical settings where the exact true number of classes is not known a priori. Finally, we find the log-normalized TURTLE model loss to be indicative of the true value of $K$, with the minimum value achieved when $K$ equals the true number of clusters. In practical settings, one can consider using this metric to select the value of $K$ when a value/proxy for the true number of clusters is not available.

Next, we perform an experiment to test whether, with datasets that are inherently hierarchical, implementing our approach can yield an advantage over flat clustering with the backbone model. To do so, we consider the INaturalist21 dataset (Van Horn et al., 2021), which contains 2.7 million images of natural species labelled at different taxonomy levels (1103 families, 273 orders, 51 classes, 13 phylums, 3 kingdoms). We train TURTLE to model clusters at the more fine-grained *family* taxonomy level ($K_{family} = 1103$). Then we implement our L2H procedure on top, and use the produced hierarchy to make clustering predictions at coarser taxonomy levels. For instance, $K_{order} - 1$ steps from the end of the procedure (see Algorithm 1), we have a $K_{order}$-clustering of the data points, and we test its performance on the test set (at the *order* taxonomy level). Then, we train instances of TURTLE at each coarse taxonomy level, i.e., $K \in \{273, 51, 13, 3\}$, and report the corresponding test set performance for comparison. Notably, training TURTLE at the fine-grained level, and using our L2H approach to construct a hierarchy and make predictions at more coarse levels, achieves better clustering performance than training separate TURTLE models at each taxonomy level. Note that this performance improvement also comes with much lower compute time, as only a single instance of the TURTLE model is trained (at the finest-grained taxonomy level). We report the results in Table 8.

| Model | Family | | | | Order | | | | Class | | | | Phylum | | | | Kingdom | | | |
|---|---|---|---|---|---|---|---|---|---|---|---|---|---|---|---|---|---|---|---|---|
| | nmi | ari | acc | lp | nmi | ari | acc | lp | nmi | ari | acc | lp | nmi | ari | acc | lp | nmi | ari | acc | lp |
| TURTLE | **0.552** | **0.052** | **0.140** | **0.373** | 0.512 | 0.075 | 0.172 | 0.562 | **0.498** | 0.101 | 0.203 | **0.827** | **0.514** | 0.244 | 0.268 | **0.893** | 0.479 | 0.461 | 0.663 | 0.873 |
| L2H-TURTLE | **0.552** | **0.052** | **0.140** | **0.373** | **0.517** | **0.097** | 0.181 | **0.572** | 0.497 | **0.123** | **0.229** | 0.797 | **0.515** | **0.294** | **0.385** | 0.877 | **0.561** | **0.562** | **0.734** | **0.920** |

Table 8: Comparison on the INaturalist21 dataset between the clustering performance of our L2H approach at coarse levels of the hierarchy, and the results obtained by training an instance of the backbone model at each coarse level. Results are averaged across five independent runs, bolding the best results and results that are statistically indistinguishable from the best.

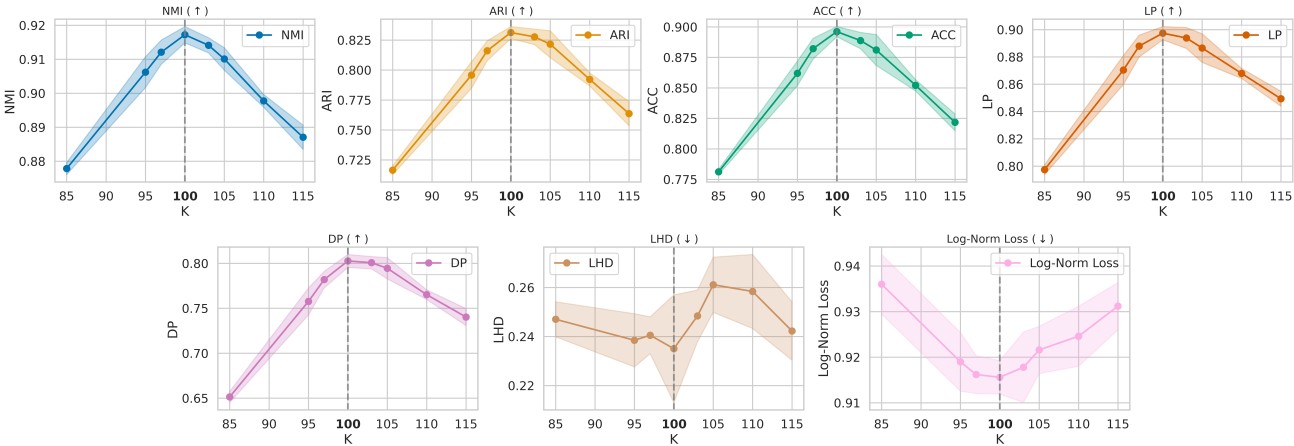

Figure 5: Sensitivity analysis for L2H-TURTLE on the CIFAR-100 dataset with respect to the $K$ hyperparameter, which corresponds to the number of leaves in the hierarchy and the number of clusters set for training the pre-trained flat model (TURTLE in this case). Note that the true number of clusters is equal to 100. Results for both flat (NMI, ARI, ACC, LP) and hierarchical (DP, LHD) metrics are included, with standard deviations across five independent runs reported as shaded areas around the line indicating mean values. We also include the log-normalized TURTLE model loss—reported in the rightmost plot in the bottom row—that proves to be indicative for model selection with respect to the $K$ hyperparameter.

Finally, we provide an ablation where we test a slightly different merging strategy for our L2H algorithm. In particular, instead of selecting the lowest-scoring group $G^* \in \mathrm{argmin}_{G \in \mathcal{G}} s(G)$ and merging it with the most related group $G^\dagger \in \mathrm{argmax}_{G \in \mathcal{G} \setminus \{G^*\}} \frac{1}{|G|} \sum_{c \in G} rp(c)$ (see details in Section 3), we directly consider all possible pairs of groups for merging, and merge the pair $G^*, G^\dagger$ that results in the highest value for $\frac{1}{|G^\dagger|} \sum_{c \in G^\dagger} rp(c)$, where $rp(c) = \sum_{\boldsymbol{x} \in \mathcal{D}^{G^*}, h_\theta^m(\boldsymbol{x}; G^*) = c} g_\theta^m(\boldsymbol{x}; G^*)$ (as also defined in Equation (2)). We report the results with this alternative merging strategy in Table 9, only reporting hierarchical clustering metrics, as the merging strategy does not impact the flat level clustering. Results show that considering all pairs of groups for merging at each step, which also results in a computational overhead, does not bring benefits in performance.

| | DP ($\uparrow$) | LHD ($\downarrow$) |
|---|---|---|
| L2H-TURTLE (alt) | 0.797 | 0.246 |
| L2H-TURTLE | 0.803 | 0.235 |

Table 9: Performance comparison on the CIFAR-100 dataset of our L2H approach, using TURTLE as backbone model, comparing the merging strategy described in Algorithm 1 with an alternative merging strategy where at each step all pairs of groups of clusters are considered for merging.

In Figures 6, 7, 8, we provide additional visualizations for the hierarchies obtained with our proposed method in our hierarchical clustering experiments, complementing the quantitative and qualitative evidence shown in Section 4.1. Note that, to produce these visualizations, as well as the other visualizations of hierarchies in this paper we used the iTOL tool (Letunic & Bork, 2024). Leaves are matched to the original labels by checking the most frequent label among data points contained in the leaf. In addition to the matched label, we report the purity of each leaf in percentage.

In Figure 9, we provide additional results for the ImageNet case study (Section 4.2) with different colorings for the inferred hierarchy, supplementing our results from Section 4.2. These visualizations show where the subtree of birds (used in Figure 3b) is located within the complete tree and in relation to other superclasses, such as mammals, reptiles, dogs, and clothing.

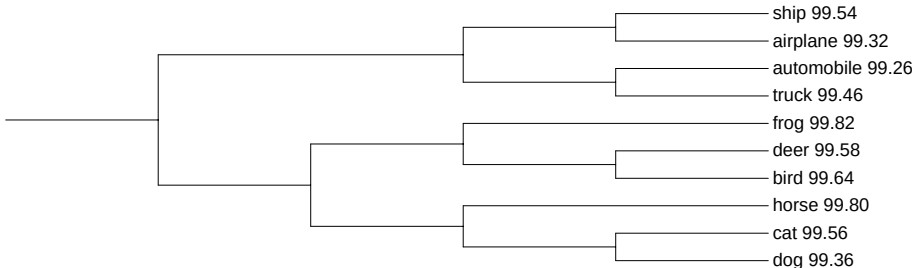

Figure 6: Visualization of the hierarchical clustering produced by L2H-TURTLE for the CIFAR-10 dataset.

Figure 7: Visualization of the hierarchical clustering produced by L2H-TURTLE for the CIFAR-100 dataset.

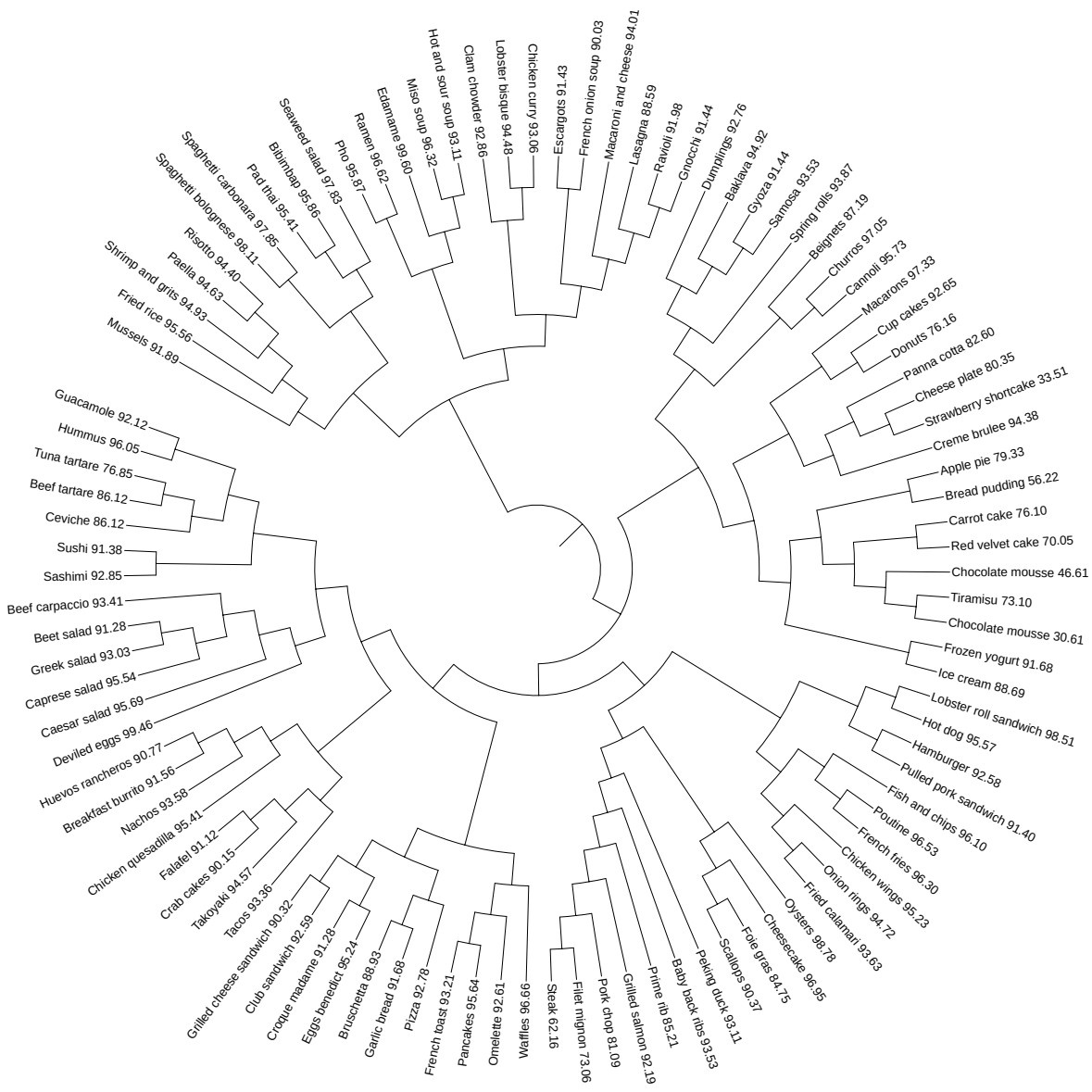

Figure 8: Visualization of the hierarchical clustering produced by L2H-TURTLE for the Food-101 dataset.

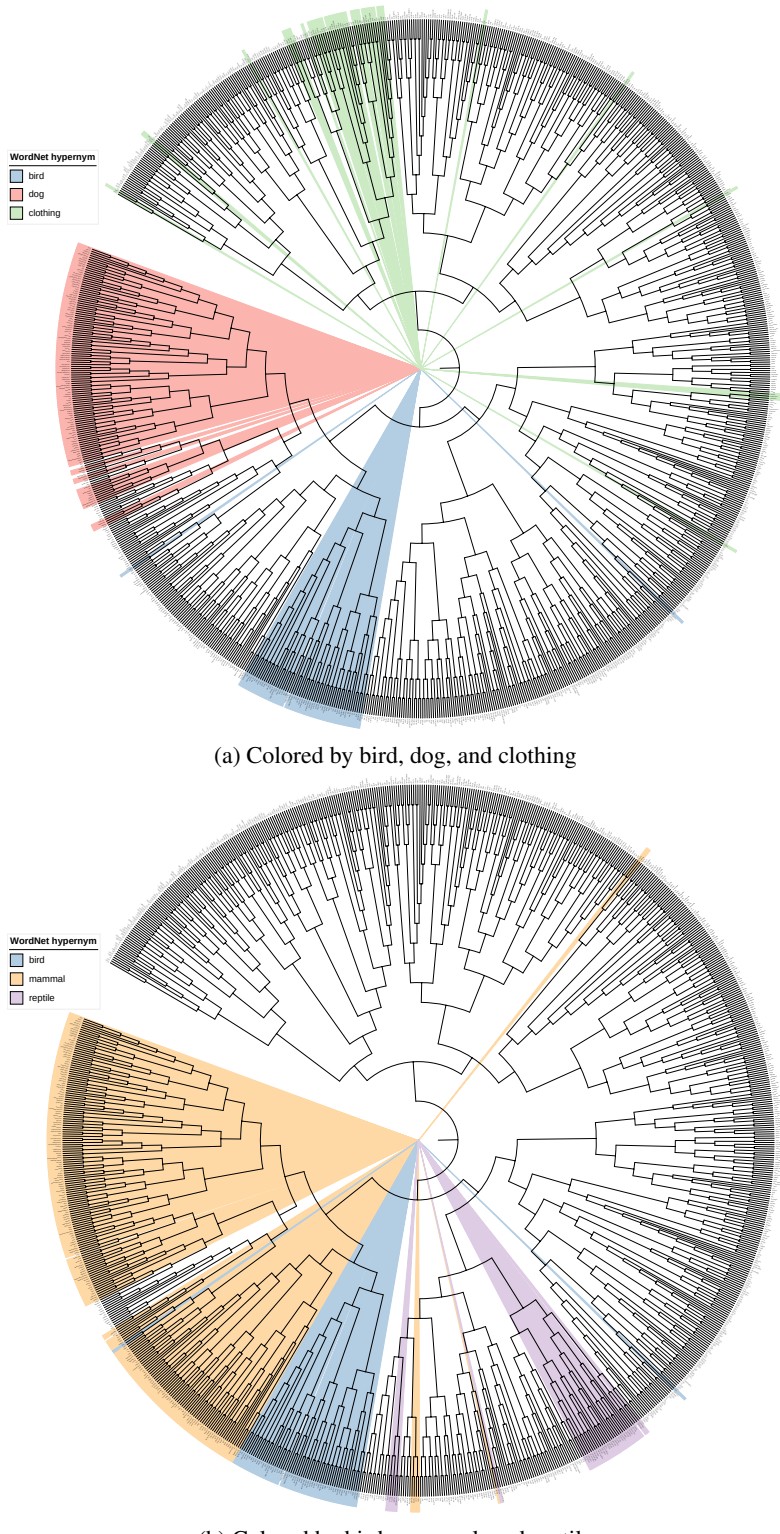

(a) Colored by bird, dog, and clothing

(b) Colored by bird, mammal, and reptile

Figure 9: Visualization of the hierarchy produced by our method in the experiment on ImageNet from Section 4.2. We show the complete tree of 1k classes colored by the corresponding WordNet hypernyms "bird", "dog", and "clothing" (Figure 9a) and by "bird", "mammal", and "reptile" (Figure 9b).

