# OpenReview forum: "From Logits to Hierarchies: Hierarchical Clustering made Simple"
_ICML.cc/2025/Conference — ICML 2025 poster_

### Official Review · Reviewer_CaXW · 2025-03-02

**Overall Recommendation:** 4

**Summary:**

A method for deriving a hierarchy of clusters from the logits from a flat clustering model, thus allowing the use the accurate leaf clustering from flat models.

##Update After Rebuttal
Most of my concerns have been addressed and I think it's a good paper. I've upgraded to a 4.

**Claims And Evidence:**

The general idea to begin with a flat clustering method and then use the logits to form a hierarchy makes sense and is well-supported by the experiments. I have some minor comments and requests.

If I understand correctly, the leaf accuracy is unaffected by the application of L2H, so the scores in Table 1 for L2H-TEMI and L2H-TURTLE, respectively. If so, I think this should be made explicit somewhere. Currently, just reading Table 1, it seems as though you are presenting results for a novel flat clustering algorithm. Perhaps they could just be called TEMI and TURTLE in the Table, and then it can be stated that your method is able to use the high leaf accuracy of SOTA flat clustering methods.

The runtime figures are stated to include the time to train TURTLE. It should also be acknowledged that TURTLE, and hence L2H-TURTLE, requires access to foundation models, which of course have a very long train time, whereas the comparison methods do not.

**Essential References Not Discussed:**

None.

**Experimental Designs Or Analyses:**

Experiments seem correct.

**Methods And Evaluation Criteria:**

The method is straightforward and effective. It is perhaps a bit simple to just take the cluster with the least confident assignments and merge it with another that has the second-highest average logits for that cluster. I wonder how the method compares to other simple operations on logits, such as computing rp(c) for all pairs and merging the highest.

**Other Comments Or Suggestions:**

line 37 RHS: "struggle to handle to" -> "struggle to handle"
line 236, LHS: "adjusted Random index" -> "adjusted Rand index"

**Other Strengths And Weaknesses:**

None.

**Questions For Authors:**

See above.

**Relation To Broader Scientific Literature:**

This algorithm could be useful as it would reduce the deep hierarchical clustering problem to the deep clustering problem.

**Theoretical Claims:**

n/a

---

> ### Author Rebuttal · Authors · 2025-04-01
>
> We thank the Reviewer for the positive feedback, in particular for praising the idea, effectiveness, and usefulness of our method. We appreciate the useful feedback/suggestions, and address the concerns below.
>
> > If I understand correctly, the leaf accuracy is unaffected by the application [...]
>
> The Reviewer is correct that our method by construction retains the leaf-level flat clustering performance of the pre-trained flat model. We make it explicit already in the text (e.g. lines 234-236, right column), but we agree with the Reviewer that in the interest of clarity it can be made more explicit in table 1. In the updated manuscript we add to the caption of Table 1 the final sentence *Notably,the application of L2H does not affect flat clustering performance, retaining the clustering performance of the pre-trained
> model (TURTLE, TEMI) at the leaf level.*, using italic to highlight it.
>
> We also perform an additional experiment on the INaturalist21 (https://github.com/visipedia/inat_comp/tree/master/2021) dataset, which we use a setting to test whether our method can bring an advantage over flat clustering methods, used as backbone, when the nature of a dataset is inherently hierarhical. The INaturalist dataset contains ~2.7million images of species labelled at different taxonomy levels (1103 families, 273 orders, 51 classes, 13 phylums, 3 kingdoms). We perform the following experiment. First we train five instances of TURTLE to model clusters at five different taxonomy levels, i.e. varying K across instances in the range $K \in \\{ 1103, 273, 51, 13, 3 \\}$. Then we apply our L2H algorithm on top TURTLE trained at the most fine-grained taxonomy level (i.e. *family*, K=1103), and use the produced hierarchy to make clustering predictions at the more coarse levels. We repeat the experiment across multiple seeds, and compare the performance of the two strategies. The results show that inferring clustering predictions at more coarse levels via the produced hierarchy with L2H leads to better performance compared to re-training TURTLE at the each corresponding level. Results at https://files.catbox.moe/0nt1uq.pdf
>
> > The runtime figures are stated to include the time [...]
>
> We have refined the manuscript accordingly to clarify this point. The Reviewer is correct, and to confirm that our approach is both more computationally efficient and more performant than deep specialized hierarchical approaches we have trained the best performing baseline (TreeVAE) on the CIFAR-100 dataset CLIP embeddings. The results prove that the performance of TreeVAE improves compared to training in data space, but it is still markedly outperformed by our approach. Note that in this setting TreeVAE takes more than 2 hours on a GPU to train, while e.g. L2H turtle takes under 2 minutes. Since all models in this comparison have access to pre-trained embeddings from foundation models, these results confirm our method is markedly more computationally efficient than alternative deep specialized hierarchical approaches. Results at https://files.catbox.moe/gz0tu8.pdf
>
> > The method is straightforward and effective.[..]
>
> Note that we motivate our choice of merging the cluster/group with the lowest score in the Rebuttal for Reviewer YXaT. We appreciate the interest/suggestion from the Reviewer and will include an ablation on the updated version of the manuscript to test the Reviewer's idea of computing rp(c) for all pairs and merging the highest. It will make an interesting ablation in our work.
> > Other Comments Or Suggestions
>
> We appreciate the Reviewer signaling these typos, which we will fix in the manuscript.
>
> ---
>
> We are happy to answer any additional questions and would appreciate it if the Reviewer would consider raising their score to full acceptance.

---

> > ### Comment · Reviewer_CaXW · 2025-04-04
> >
> > Thanks for the reply and clarifications.
> >
> > The additional results comparing to seem to retraining TURTLE at each level are helpful, and I would suggest including them in the paper. I am not sure why the results comparing to TreeVAE use CLIP instead of TURTLE for features? I would think the best comparison is them all having the same backbone.

---

> > > ### Author Response · Authors · 2025-04-07
> > >
> > > We thank the Reviewer for the feedback, and are glad that the results comparing the performance of our method with the performance obtained retraining TURTLE at different levels are helpful.  We will include these results in the updated version of the manuscript.
> > >
> > > To address the Reviewer's question, we'd like to clarify that TURTLE is a flat clustering model (as it is TEMI), and hence there's no natural way to integrate it as backbone of TreeVAE. We compare with TreeVAE trained on CLIP embeddings to provide a comparison where all models (TreeVAE, L2H-TEMI, L2H-TURTLE) make use of CLIP representations. Our approach proves to achieve markedly better performance, with substantially higher computational efficiency.
> > >
> > > We hope to have addressed the Reviewer's question, and would appreciate it if they'd consider raising their score to a full acceptance.

---

### Official Review · Reviewer_pCAf · 2025-03-07

**Overall Recommendation:** 3

**Summary:**

The paper introduces Logits to Hierarchies (L2H), a novel hierarchical clustering method that uses pre-trained non-hierarchical clustering models to build hierarchical structures. L2H is lightweight, doesn't require fine-tuning, and uses logits to generate clusters, outperforming existing deep hierarchical clustering methods in both performance and efficiency. It is also extended to supervised settings, showing its ability to recover meaningful hierarchies from pre-trained classifiers like ImageNet. Empirical results on CIFAR-10, CIFAR-100, Food-101, and ImageNet demonstrate L2H's potential for interpretability and bias detection in supervised models.

**Claims And Evidence:**

The introduction of the proposed method in the paper is clear and well-structured. The authors provide a detailed explanation of their Logits to Hierarchies (L2H) approach, including its motivation, algorithmic procedure, and mathematical formulation. Additionally, they present pseudocode and visual illustrations that facilitate understanding

However, the experimental results reveal a significant performance gap between L2H and the deep hierarchical clustering methods (e.g., DeepECT, TreeVAE). The credibility of the experimental results requires further scrutiny. Specifically, the validity and fairness of the baseline implementations, as well as the generalizability of L2H across different datasets and real-world applications, should be carefully examined.

**Essential References Not Discussed:**

The paper covers the relevant literature, and no essential references appear to be missing.

**Experimental Designs Or Analyses:**

The experimental design primarily focuses on evaluating L2H against deep hierarchical clustering models (DeepECT, TreeVAE) on CIFAR-10, CIFAR-100, and Food-101, with additional results on ImageNet-1K. While the methodology appears reasonable, there are concerns regarding the validity of comparisons:

1.	The reported performance gap between L2H and baseline methods is significant, yet the paper lacks details on whether baselines were optimally configured. Ensuring fair hyperparameter tuning and implementation details is essential for a valid comparison.

2.	While the paper claims scalability, the chosen datasets may not fully reflect real-world large-scale clustering challenges. Further analysis on more complex, high-dimensional datasets would strengthen the validation.

3.	The authors highlight L2H’s efficiency, but more transparency on runtime conditions (e.g., hardware specifications, dataset size variations) is needed to verify the claimed advantages.

**Methods And Evaluation Criteria:**

The proposed method, L2H, is well-suited for the problem of hierarchical clustering. It builds on pre-trained flat clustering models and uses logits to construct hierarchies, which is a novel approach. The evaluation criteria are appropriate, with metrics that assess both the quality of the flat clustering (e.g., NMI, ARI, Accuracy) and the hierarchical structure (e.g., Dendrogram Purity, Least Hierarchical Distance). The authors also provide a case study on ImageNet, demonstrating the method's applicability to supervised settings, which adds to the method's generality and practical relevance.

However, the authors claim that L2H is scalable to large-scale datasets and computationally efficient. The datasets used (CIFAR-10, CIFAR-100, Food-101) may not be sufficient to fully validate its scalability in real-world scenarios.

Additionally, the comparison with baselines such as DeepECT and TreeVAE requires clarification. Given the large performance gap, it is unclear whether these baselines were optimally configured. More details on hyperparameters, training settings, and computational budgets would ensure a fair comparison.

**Other Comments Or Suggestions:**

I have no other comment.

**Other Strengths And Weaknesses:**

**Strengths:**

The proposed method is scalable and efficient, making it suitable for large-scale datasets.

The paper provides some empirical evidence, including visualizations and ablation studies, to support the claims.

The extension to supervised settings adds practical relevance, particularly for interpretability and bias detection.

**Weaknesses:**

The method relies on pre-trained models, which may limit its applicability in scenarios where such models are not available.

While the method is efficient, the quality of the hierarchy depends on the quality of the pre-trained model's logits, which could be a limitation if the pre-trained model performs poorly.

In this paper, the validity of the method is verified only by experiment, and the theory is lacking.

The paper claims the method is lightweight and scalable but lacks extensive experiments on extremely large datasets (e.g., millions of samples or hundreds of thousands of classes). Experiments are limited to smaller datasets like CIFAR and Food-101, leaving scalability claims unverified for the so-called large-scale data.

**Questions For Authors:**

1.	The method relies on pre-trained models. Have the authors explored scenarios where pre-trained models are not available, or where the pre-trained model's performance is suboptimal? How does L2H perform in such cases?

2.	This paper only uses experiments to verify the effectiveness of the method. Is there any relevant theoretical guarantee to illustrate the effectiveness of the method?

3.	The TEMI employs CLIPViTL/14 representations of the data, while TURTLE employs both CLIPViTL/14 and DINOv2 ViT-g/14 representations. Are the compared methods utilize these representations?

**Relation To Broader Scientific Literature:**

The paper is well-situated within the broader literature on hierarchical clustering. The authors discuss the limitations of existing deep hierarchical clustering methods and position their work as a more scalable and efficient alternative. They also connect their work to the growing interest in interpretability and bias detection in machine learning models, particularly in supervised settings. The paper builds on recent advances in pre-trained models and logit-based approaches.

**Theoretical Claims:**

The paper does not present any theoretical proofs or claims.

---

### Official Review · Reviewer_YXaT · 2025-03-13

**Overall Recommendation:** 3

**Summary:**

- Main algorithmic/conceptual ideas: Using logits of flat clustering model, get hierarchical structure on top of them.
- Main findings/results: deep hierarchical clustering methods have low clustering performance and slow runtime; learned hierarchies from image datasets are interpretable.

**Claims And Evidence:**

Yes in general. The whole paper is overall clear.

**Essential References Not Discussed:**

Minor: for the benefits of modeling a hierarchy in the data are not restricted to the unsupervised setup, there are some works to use hierarchical data for out-of-distribution detection and the hierarchy is used for supervised learning.

[1] Khrulkov, Valentin, et al. "Hyperbolic image embeddings." Proceedings of the IEEE/CVF conference on computer vision and pattern recognition. 2020.

[2] Linderman, Randolph, et al. "Fine-grain inference on out-of-distribution data with hierarchical classification." Conference on Lifelong Learning Agents. PMLR, 2023.

[3] Sinha, Aditya, et al. "Learning Structured Representations with Hyperbolic Embeddings." Advances in Neural Information Processing Systems 37 (2024): 91220-91259.

**Experimental Designs Or Analyses:**

Questions:
- Is it possible to show non-hierarchical clustering metrics for TEMI and TURTLE? My guess is your method will have an absolutely better performance with a stronger flat clustering initialization, especially when I noticed that TEMI and TURTLE are based on CLIP embeddings which is quite powerful (line 765-768). Therefore, it's better to show this increment with your method as an important ablation study. Your results will be stronger if you show there is a significant increase with your method on top of any flat clustering methods.
- Out of curiosity, why are flat deep clustering methods faster? Is this true for all (maybe SOTA) deep clustering methods?
- Going back to the choice of flat clustering method, if you use CLIP based embeddings, in Table 2, for the runtime to train the TURTLE model, I would imagine you didn't include the time to train CLIP, which sounds unfair for other methods.
- Additionally, there's another concern about interpretability arguments in your visualization, because basically CLIP has already seen text embeddings so it could infer some knowledge directly from text, thus probably having an additional advantage to match with wordnet.
- But overall I don't think this will be a significant issue if the author can either provide reasonable explanation on the choice of clustering method, or showing the result of your method + other weaker flat clusteirng method as initialization. I understand that the final metrics probably won't be as good as what you reported now, but that is still valuable to the community and will make your paper much more convincing.

**Methods And Evaluation Criteria:**

Questions for methods:
- line 131: how do you decide the number of clusters $K$? Is this decided by your input flat clustering method?
- line 199: why do you choose the lowest score (predicted probabilities) for merging, say when the aggregation function is summation, or others?
- line 144: Seems like your method is based on the assumption that cluster logits (or probabilities) are a proxy to measure cluster similarities. Why is this assumption reasonable? When can this assumption fail? I encourage the authors to briefly discuss on this point.

Metrics and datasets look good to me, both are reasonable for clustering evaluation.

**Other Comments Or Suggestions:**

Can you explain Figure 1 iteration 2? This is not contained in the caption and why is the pink box merged with the first two boxes? Also what does the top bar for iteration 2 mean where it contains both blue and yellow parts?

**Other Strengths And Weaknesses:**

- Originality: Hard to judge based on my familiarity of the literature.
- Significance: The proposed method is lightweight and empirically works well. Although see "Experimental Designs Or Analyses" for concerns of the experiment section that may mislead the interpretation of the results.
- Clarity: overall well-written paper.

**Questions For Authors:**

For detailed questions see comments above. Here let me summarize major questions:
- selection of initial clustering methods in your experiments
- why choose the lowest score for merging

I would like to raise my score if the author can elaborate on these questions.

**Relation To Broader Scientific Literature:**

This can be categorized into a bottom-up approach of hierarchical clustering, and is built on trained non-hierarchical clustering methods.

**Theoretical Claims:**

N/A

---

> ### Author Rebuttal · Authors · 2025-04-01
>
> We thank the Reviewer for praising performance and efficiency of our method, our metrics/datasets, as well as the clarity of our paper. We appreciate useful the feedback and suggestions, and address the concerns below.
> > line 131[..]
>
> The number of leafs in the hierarchy obtained with our approach is the number of clusters $K$ modelled by the pre-trained flat model, which is a hyperparameter. Note that we show our approach is robust to changes in $K$, and should an a-priori value for $K$ not be available, the loss of the flat model (e.g. TURTLE) can be useful signal to set it (See rebuttal to Reviewer CKwm).
> > line 199 [..]
>
> The score we define captures the aggregated probability mass that the model assigns to each cluster/group. When a cluster/group receives the lowest score, depending on the exact aggregation, it indicates one of two scenarios (or both):
>
> - It contains fewer samples, suggesting it is more specialized than others.
> - The model is less confident when assigning samples to it, often because it's highly similar to another cluster/group.
>
> In either case, the cluster/group is a natural candidate for early merging in a bottom-up hierarchy from fine-grained to coarse. In the first case, it is more specialized than others. The second case is a little more subtle. Consider the case of overclustering, where the number of leaf nodes K is larger than the true number of clusters. In this case, redundant clusters modelling the same true class present low-confidence assignments, as the model can't clearly distinguish between them when assigning samples. This low confidence reduces their score, causing them to be merged early, effectively correcting the overclustering. Thus, our score guides the hierarchy-building process in a principled way, encouraging early merging of either highly specific or redundant clusters, ultimately leading to an accurate hierarchy, as validated empirically.
> > line 144: [..]
>
> We agree that this assumption underlies our method. While empirically supported, we acknowledge that this assumption may not always hold, for instance in case the flat model is poorly calibrated. We have updated the manuscript to explicitly discuss this assumption and its potential limitations.
> > Is it possible to [..]
>
> As we state in lines 234 - 236 (right column), by construction, our method retains the clustering performance of the pre-trained model at the leaf level, hence non-hierarchical clustering metrics for TEMI/TURTLE match the L2H-TEMI/L2H-TURTLE results in Table 1. However, our results on the INaturalist21 dataset (see rebuttal to Rev. CaXW) prove that our approach can be used to model multiple granularity levels in datasets with a hierarchical structure. Notably, our approach surpasses the performance obtained by training multiple instances of a flat model (e.g. TURTLE), one at each given granularity. In such a setting, our hierarchical approach surpasses the flat model (e.g. TURTLE) on flat metrics at all but the finest granularity (where the performance matches).
> > Out of curiosity [..]
>
> It is not necessarily true in principle that all flat clustering methods are faster than hierarchical methods. However, there's been a lot recent research on highly performant and efficient flat clustering (e.g TURTLE), while a comparable research effort has not been witnessed for hierarchical models.
> > Going back to the choice [..]
>
> See rebuttal to Rev. CaXW.
> > Additionally [..]
>
> Note that in section 4.2 we use InternImage as a backbone mode - not based on CLIP embeddings.
> > But overall [...]
>
> We appreciate the suggestion from the Reviewer, and implement our method on top of the TCL flat clustering model [1] on the datasets used in Table 1. TCL (i) does not rely on pre-trained embeddings, and (ii) achieves weaker flat results than TEMI/TURTLE. Results at https://files.catbox.moe/cfufvm.pdf. Still, with our approach (L2H-TCL) we outperform deep hierarchical models (DeepECT,TreeVAE) across all flat/hierarhical metrics, which strenghtens our contribution.
> [1] Li et al. Twin Contrastive Learning for Online Clustering, IJCAI, 2021.
> > Can you explain Figure 1[..]
>
> In the second step in Fig.1 the pink cluster, selected for merging, is merged with the group containing yellow and blue clusters, as this group has the most reassigned predicted probability mass. The bar contains blue/yellow parts to represent that it aggregates the probability mass reassigned to blue/yellow clusters, that were grouped together at the previous step.
> > Minor: for the benefits of modeling [..]
>
> We agree with the Reviewer that the benefits of modelling a hierarchy are not restricted to the unsupervised setup, and show results to support this point in section 4.2. We appreciate the useful references and have integrated them in the updated manuscript.
>
> We are happy to answer any additional questions, and would appreciate it if the Reviewer would consider increasing their score to an acceptance.

---

> > ### Comment · Reviewer_YXaT · 2025-04-04
> >
> > I thank the author for your detailed rebuttal. I think additional comparative experiments strengthens your paper and I strongly recommend the author to add these new experiments into the updated version.
> >
> > Therefore I will raise my score to 3.

---

> > > ### Author Response · Authors · 2025-04-08
> > >
> > > We thank the Reviewer for raising their score to an acceptance, and for the useful feedback. We will include the additional experimental results from the rebuttal in the updated version of the manuscript.

---

### Official Review · Reviewer_CKwm · 2025-03-13

**Overall Recommendation:** 3

**Summary:**

This paper addresses the issues of traditional hierarchical clustering algorithms, such as high computational costs, and strong model dependency. It proposes a lightweight hierarchical clustering algorithm that directly constructs hierarchical structures using the logits output by pre-trained models, enabling multi-granularity clustering without fine-tuning. Furthermore, extensive experiments demonstrated the effectiveness and generalizability of the proposed method.

**Claims And Evidence:**

The author conducted several experiments to demonstrate the effectiveness of the proposed method. However, increasing the number of datasets and comparison methods would enhance the credibility of the results.

**Essential References Not Discussed:**

Given the related work on hierarchical clustering presented at ICML 2024, it is recommended that a performance comparison analysis with the proposed method be conducted. This would provide a more comprehensive evaluation and highlight the new approach's advantages and limitations in the context of recent advancements.

**Experimental Designs Or Analyses:**

The inspection has been carried out, and no issues were found.

**Methods And Evaluation Criteria:**

Yes.

**Other Comments Or Suggestions:**

Considering the aforementioned Weaknesses.

**Other Strengths And Weaknesses:**

Strengths:
1. The proposed method can process ImageNet-scale data in just a few minutes on a CPU, significantly outperforming deep methods such as TreeVAE.
2. The author conducted experiments to demonstrate the generalizability of the proposed method.

Weaknesses:
1. Theoretically, the design basis of the masked Softmax lacks rigorous mathematical derivation, and the relationship between the hierarchical clustering objective function and the downstream tasks is not clearly established.
2. In the evaluation metrics, only traditional clustering metrics were used, and there is a lack of metrics specifically tailored for hierarchical clustering.
3. No sensitivity analysis of the parameters was conducted, such as the depth of the tree.
4. The summary of contributions and the future work section need further refinement.
5. The majority of the references are from five years ago, and there is a need to incorporate more recent and advanced studies. This is important because citing newer literature ensures that the research is aligned with current trends and advancements in the field.

**Questions For Authors:**

1. Does the appendix introduce four datasets, but only three datasets were actually used in the experiments?
2. Table 3 does not provide comparisons with other methods, making it difficult to determine the effectiveness of the proposed approach.
3. Should Table 5 be moved to the main text? The appendix should be revised to remove any redundant information.
4. The remaining issues can be referred to the section on Weaknesses.

**Relation To Broader Scientific Literature:**

The paper's key contributions address well-known challenges in hierarchical clustering, such as computational cost and model dependency. They build upon prior work by leveraging pre-trained models without fine-tuning, thereby advancing the state-of-the-art inefficient and semantic-aware clustering methods.

**Theoretical Claims:**

The inspection has been carried out, and no issues were found.

---

> ### Author Rebuttal · Authors · 2025-04-01
>
> We thank the Reviewer for the valuable feedback and suggestions. We appreciate the praise for the efficiency and efficacy of our method, and the highlighting of its value in addressing well-known challenges in hierarchical clustering. We address concerns/questions below.
> > However, increasing the number of datasets and comparison methods [..]
>
> See comparison with [1] below +  other Rebuttals (e.g. L2H+TCL and TreeVAE+CLIP comparisons, INaturalist results).
> > Given the related work on [...]
>
> We assume the Reviewer refers to this ICML 2024 workshop paper[1], on zero-shot flat clustering. The proposed strategy (UMAP dim.red. on embeddings + Ward's agglomerative clustering) incurs in high computational cost for large datasets, and more importantly *does not allow for inference on unseen data*. Hence, to test the method from Lowe et al. for comparison we necessarily both train and evaluate it on the test set, thereby giving it an advantage in seeing test data for training. Despite this, this method still underperforms compared to our proposed approach, which further validates and contextualises the effectiveness of our method. Results on CIFAR-100 at https://files.catbox.moe/4nhqck.pdf
> > Theoretically, the design basis of the masked Softmax [..]
>
> We introduce the masked softmax as a principled variation of the softmax function that allows masking in the indexes. Note that our function definition ensures that a valid probability distribution over the unmasked elements is produced. We do not consider a downstream task in our hierarchical clustering experiments. We would thus appreciate it if the Reviewer could provide clarifications regarding the comment "the relationship between the hierarchical clustering objective function and the downstream tasks is not clearly established".
> > In the evaluation metrics, only traditional clustering metrics were used[..]
>
> Note that we use Dendrogram Purity (DP)[2,3] and Least Hierarchical Distance (LHD) - both metrics tailored for hierarchical clustering.
> > No sensitivity analysis of the parameters was conducted, such as the depth of the tree.
>
> We provide a sensitivity analysis of L2H-TURTLE on the CIFAR-100 dataset, with respect to the depth of the tree. The depth of the tree is controlled by the $K$ hyperparameter corresponding to the number of clusters modelled by the flat model and therefore to the number of leafs in the tree. The analysis demonstrates that across all hierarchical and flat metrics the best performance is achieved when K equals the true number of classes (100). When $K$ deviates from the true number of clusteris, performance degrades gracefully and a meaningful hierarchy is still recovered. This robustness is particularly important in practical setting where the true number of clusters is not known a-priori; in such cases the log-normalized TURTLE model loss can provide useful signal to select the hyperparameter $K$. Results at https://files.catbox.moe/iisaqn.pdf.
> > The summary of contributions and the future work section need further refinement.
>
> We appreciate the suggestion from the Reviewer, and have refined the manuscript accordingly, summarizing and highlighting more clearly the contributions of our work. As well, we have deepened the section on future work, also in light of some additional results shown in this rebuttal.
> > The majority of the references are from [...]
>
> We share the Reviewer’s view on the importance of citing recent literature, and have made our best effort to do so throughout the paper. However, as noted in the recent relevant work[2], the number of deep learning-based approaches proposed in the last few years to address hierarchical clustering remains surprisingly limited. This underlines the relevance and timing of our work, where we aim to revive the interest in this underexplored area, building on recent advancements (e.g. in flat clustering models). To that end, we compare with the most recent and relevant baselines[2] and build upon SOTA flat clustering models. Finally note that we have followed the recommendation from the Reviewer contextualizing our approach in comparison with [1] above.
>
> **Replies to "Questions for the Authors"**
>
> 1.We introduce CIFAR-10, CIFAR-100, Food-101 that we used in section 4.1 and ImageNet1k that we used in Section 4.2 and Table 3.
>
> 2.In line 753 we highlight that comparisons with baselines (e.g., DeepECT, TreeVAE) are not feasible on ImageNet1k, as these methods lack the scalability to handle datasets of this magnitude/complexity.
>
> 3.We reported Table 5 in the Appendix as it consists of an ablation validating the stability wrt the design choice of the aggregation function. We appreciate the Reviewer's suggestion and have refined the Appendix by removing redundant results.
>
> [1] Lowe et al.(2024)
> [2] Manduchi et al.(2023)
> [3] Kobren et al al.(2017)
>
> We are happy to answer any additional questions, and would appreciate it if the Reviewer would consider increasing their score to an acceptance.

---

> > ### Comment · Reviewer_CKwm · 2025-04-07
> >
> > The authors have adequately addressed the major concerns raised in the previous review. Based on the improvements and clarifications provided in the rebuttal, I am raising my score to 3.

---

> > > ### Author Response · Authors · 2025-04-08
> > >
> > > We thank the Reviewer for raising their score to an acceptance. We are glad to have been able to address their concerns, and are grateful for the useful feedback.

---

### Decision · Program_Chairs · 2025-05-01

**Decision:**

Accept (poster)

**Comment:**

This paper proposes a lightweight method for hierarchical clustering that leverages pre-trained non-hierarchical clustering models, outperforming specialized deep hierarchical clustering models in terms of efficiency, scalability, and performance, and demonstrating its applicability in both unsupervised and supervised settings. After the rebuttal, it receives two weak accept and one accept. Its merits, including the interesting idea, very good efficiency on large dataset ImageNet-1K, and good results, are well recognized by the reviewers. The response well addresses the reviewers' concerns about the additional comparison, detailed analysis, and so on. I think the current manuscript meets the requirement of this top conference and recommend for acceptance. Please incorporate the revision in the updated manuscript.